# Robust and Diverse Multi-Agent Learning via Rational Policy Gradient

**Niklas Lauffer**[1]   **Ameesh Shah**[1]   **Micah Carroll**[1]
**Sanjit A. Seshia**[1]   **Stuart Russell**[1]   **Michael Dennis**[2]
[1]UC Berkeley    [2]Google Deepmind
nlauffer@berkeley.edu

## Abstract

Adversarial optimization algorithms that explicitly search for flaws in agents' policies have been successfully applied to finding robust and diverse policies in multi-agent settings. However, the success of adversarial optimization has been largely limited to zero-sum settings because its naive application in cooperative settings leads to a critical failure mode: agents are irrationally incentivized to *self-sabotage*, blocking the completion of tasks and halting further learning. To address this, we introduce *Rationality-preserving Policy Optimization (RPO)*, a formalism for adversarial optimization that avoids self-sabotage by ensuring agents remain *rational*—that is, their policies are optimal with respect to some possible partner policy. To solve RPO, we develop *Rational Policy Gradient (RPG)*, which trains agents to maximize their own reward in a modified version of the original game in which we use *opponent shaping* techniques to optimize the adversarial objective. RPG enables us to extend a variety of existing adversarial optimization algorithms that, no longer subject to the limitations of self-sabotage, can find adversarial examples, improve robustness and adaptability, and learn diverse policies. We empirically validate that our approach achieves strong performance in several popular cooperative and general-sum environments. Our project page can be found at rational-policy-gradient.github.io[1].

## 1   Introduction

A longstanding challenge in the field of multi-agent reinforcement learning (MARL) is that of learning *robust* behavior: individual agents should be able to adapt to a variety of different strategies that other agents might exhibit. One way to achieve robustness is by training agents to iteratively find and fix flaws in their policy. In zero-sum settings, this can be naturally achieved through *self-play* [Samuel, 1959, Silver et al., 2016], where agents train against copies of themselves. Due to the adversarial nature of zero-sum self-play, agents will continually be encouraged to find new ways of attacking their opponents which will naturally lead to iterative improvement and robustification. In *general-sum* (especially cooperative) settings, however, self-play will explicitly *avoid* the weaknesses of other players, as it is harmful to the shared reward, resulting in brittle agents [Carroll et al., 2019].

Inspired by its success in zero-sum settings, we leverage a form of *adversarial optimization* (i.e., we incentivize minimizing other players' rewards) to train agents to find and fix flaws in general-sum settings. However, seeking to minimize others' rewards in cooperative settings, where all agents aim to maximize a shared reward, unsurprisingly leads to *self-sabotaging* behavior [Cui et al., 2023]. If an agent is solely incentivized to minimize the rewards of another player, the adversary can simply learn to refuse to collaborate with that teammate. Even worse, the adversary has an incentive to act irrationally by actively sabotaging its teammate's (and by extension, its own) reward, preventing meaningful learning.

---

[1]Code can be found at github.com/niklaslauffer/rational-policy-gradient.

39th Conference on Neural Information Processing Systems (NeurIPS 2025).

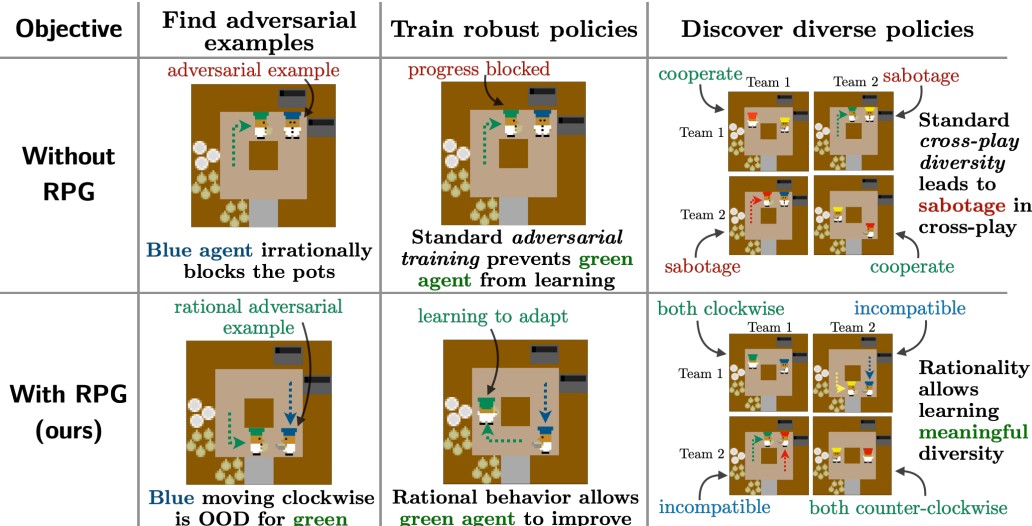

Figure 1: Rational policy gradient (RPG) allows finding rational adversarial examples, robustifying behavior, and discovering diverse policies.

In order to reap the benefits of adversarial optimization without incurring self-sabotaging behavior, we establish a new paradigm for adversarial optimization called *Rationality-preserving Policy Optimization* (RPO). We formalize RPO as an adversarial optimization problem that requires the policy to be optimal with respect to at least one policy that other agent(s) might play. This can be thought of as enforcing the agent to be *rational*: the agent must be utility-maximizing for some choice of teammates.

The rationality constraint imposed by RPO is difficult to directly integrate into a single optimization objective. To solve RPO, we introduce a novel approach called *rational policy gradient (RPG)*, which provides a gradient-based method for ensuring rational learning while optimizing an adversarial objective. RPG introduces a new set of agents called *manipulators*, one for each of the agents in the original optimization problem (which we call *base agents*). In RPG, the base agents only train to maximize their own reward in a copy of the game (called its *manipulator environment*) with their teammates replaced by their manipulator counterparts – this ensures that the base agents are solely learning to be rational. Each manipulator uses opponent-shaping [Foerster et al., 2017] to manipulate the base agents' learning and guide them towards policies that optimize the adversarial objective (e.g., achieving low reward with one another in the original *base environment*). The manipulators are discarded after training and the trained base agents give the solution to the RPO-version of the adversarial objective – whether that be related to robustness, diversity, or some other objective.

As summarized in Figure 1, we use RPG to extend several existing adversarial optimization algorithms to find adversarial examples in pretrained policies, train more robust strategies, and fully eliminate self-sabotaging behavior exhibited by existing cross-play-based diversity algorithms – an open problem in existing literature [Charakorn et al., 2023, Cui et al., 2023, Sarkar et al., 2024]. We summarize our contributions:

- We introduce a formalism called RPO that overcomes the issue of self-sabotage in any adversarial optimization algorithm.
- We introduce a gradient-based deep learning algorithm called RPG that solves RPO.
- We use RPG to construct five novel adversarial optimization algorithms that find rational adversarial examples, train robust agents, and learn diverse policies.
- We empirically demonstrate that our algorithms avoid self-sabotage and outperform existing baselines in popular cooperative environments.

## 2 Preliminaries

We consider the setting in which agents play in a *general-sum partially-observable stochastic game* $(M, \mathcal{S}, \mathcal{A}, R, \gamma, P, \Omega, \mathcal{O})$ defined as follows. $M = \{1, \ldots, k\}$ is a set of agents. $\mathcal{S}$ is a set of states with initial distribution $\mathcal{S}_0$. $\mathcal{A} = \mathcal{A}_1 \times \cdots \times \mathcal{A}_m$ is the space of joint actions; for ease

of notation, we assume without loss of generality that the action set is identical across agents. $R_i : \mathcal{S} \times \mathcal{A} \times \mathcal{S} \to \mathbb{R}$ is agent $i$'s reward function. $\gamma$ is the discount factor. $P : \mathcal{S} \times \mathcal{A} \times \mathcal{S} \to [0, 1]$ is the transition function where $\sum_{s' \in \mathcal{S}} P(s, a, s') = 1$. $\Omega = \Omega_1 \times \cdots \times \Omega_m$ is the joint observation space. $\mathcal{O} : \mathcal{S} \times \mathcal{A} \times \Omega \to [0, 1]$ is the observation function.

For sake of exposition, we limit ourselves to games with two players. Agents follow *stochastic Markovian policies*: the policy $\pi_i$ agent $i$ specifies a distribution over actions $\mathcal{A}_i$ for every observation in $\Omega_i$. Let $\Pi$ denote the joint stochastic Markovian policy space for all agents, $\Pi_i$ denote the policy space of agent $i$, and $\Pi_{-i}$ denote the *co-policy* space, the joint policy space of the co-players (all agents other than $i$). We note that our methods will work for history-dependent policies too (e.g., by allowing the policies to maintain external memory). We use $U_i(\pi_i, \pi_{-i})$ to denote the expected (over stochasticity from the environment and policy) sum of discounted rewards $\mathbb{E}\left[\sum_t \gamma^t R_i(s_t, a_t, s_{t+1})\right]$ for agent $i$ from the initial distribution over states. The *best-response* function for agent $i$ is the set-valued function $\mathrm{BR}_i : \Pi_{-i} \to \mathcal{P}(\Pi_i)$ such that $\mathrm{BR}_i(\pi_{-i}) = \arg\max_{\pi_i} U_i(\pi_i, \pi_{-i})$, denoting the set of policies agent $i$ could play to maximize the sum of discounted returns in response to $\pi_{-i}$. We investigate the problem setting in which arbitrary optimization objectives $O_i$ are given for each agent $i$ as discussed in the following section.

# 3 Rationality-Preserving Policy Optimization

## 3.1 Adversarial Optimization Causes Self-Sabotage

We call an optimization objective *adversarial* when some of the agents are explicitly incentivized to minimize the reward of another agent. For example, the adversarial training (AT) optimization problem [Gleave et al., 2019] is defined for a victim $\pi_{\text{victim}}$ and an adversary $\pi_{\text{adversary}}$ in a 2-player game. The objective for the adversary is $\min_{\pi_{\text{adversary}}} U_{\text{victim}}(\pi_{\text{victim}}, \pi_{\text{adversary}})$ while the objective for the victim is non-adversarial: $\max_{\pi_{\text{victim}}} U_{\text{victim}}(\pi_{\text{victim}}, \pi_{\text{adversary}})$. In zero-sum settings, AT has been used to train the adversary to find adversarial examples that expose robustness flaws in the victim's policy and have the victim learn to fix them.

In an attempt to train an agent that is robust in a cooperative setting, consider applying AT to the game between two players defined by the matrix in Figure 2. Suppose the victim is the row player and the adversary is the column player. Under the objectives specified by AT, no matter which policy $\pi_{\text{victim}}$ plays, the adversary will *self-sabotage* the game by playing policy

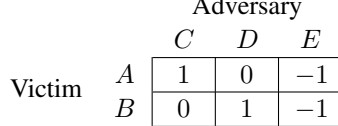

|        |     | C | D | E |
|--------|-----|---|---|----|
| Victim | A | 1 | 0 | −1 |
|        | B | 0 | 1 | −1 |

Figure 2: A cooperative game.

$\pi_{\text{adversary}}^{\text{AT}} = E$ since this automatically minimizes $\pi_{\text{victim}}$'s reward at a value of -1. Notice how $\pi_{\text{adversary}}^{\text{AT}}$ will not help $\pi_{\text{victim}}$ improve its robustness: no policy available to $\pi_{\text{victim}}$ can do well if $\pi_{\text{adversary}}^{\text{AT}}$ plays action $E$. Motivated by such examples, we define self-sabotage as follows.

**Definition 3.1** (Self-sabotage). Optimization problems in multi-agent games often include objectives distinct from the incentives in the underlying game (i.e., agents' incentive to maximize their individual reward). When the optimization problem results in policies with *irrational behavior*, behavior that causes an agent to act against its own incentive in the underlying game, we call this *self-sabotage*.

## 3.2 The Rationality-Preserving Policy Optimization Formalism

Rationality-preserving policy optimization (RPO) fixes the issue of self-sabotage in adversarial optimization problems by requiring an adversarial agent's policy to be rational, i.e., a best-response to at least one possible co-policy.

**Definition 3.2** (Rationality-preserving Policy Optimization (RPO)). For each agent $i$, let $O_i(\pi_1, \ldots, \pi_m)$ denote its adversarial optimization objective. The RPO-version of agent $i$'s objective is given by

$$\max_{\pi_i} O_i(\pi_1, \ldots, \pi_m) \quad \text{subject to } \exists \pi'_{-i} \in \Pi_{-i} \text{ s.t. } \pi_i \in \mathrm{BR}(\pi'_{-i}). \tag{1}$$

Let us walk through an example of applying RPO to the adversarial training (AT) objective $\min_{\pi_{\text{adversary}}} U_{\text{victim}}(\pi_{\text{victim}}, \pi_{\text{adversary}})$. RPO will modify AT by requiring that $\pi_{\text{adversary}}$ is rational by serving as a best response to some co-policy. Notice that the RPO constraint has no affect on $\pi_{\text{victim}}$ since it is already optimizing to be rational.

Now, if $\pi_{\text{victim}} = A$, the constraint from Theorem 3.2 will lead the adversary to find the policy $\pi_{\text{adversary}}^{\text{RPO}} = D$ since $D$ is a valid best response if the victim plays $B$. Note that while playing $E$ minimizes the adversarial objective further than playing $D$, it does not satisfy the constraint of being a rational action. Importantly, the adversary is teaching the victim something actionable: increasing their likelihood of playing $B$ would improve their reward. Moreover, the only equilibrium of this new optimization problem is for the victim to play a uniform mixture over actions $A$ and $B$. This makes the victim robust because the minimum expected reward it will obtain against any rational co-policy is $0.5$.

Notice that the AT-RPG is a strict generalization of AT since they are identical in zero-sum games. In the zero-sum setting, letting $\pi' = \pi_{\text{victim}}$ trivially satisfies the rationality constraint since $\text{BR}(\pi_{\text{victim}}) = \max_{\pi_{\text{adversary}}} U_{\text{adversary}}(\pi_{\text{victim}}, \pi_{\text{adversary}}) = \min_{\pi_{\text{adversary}}} U_{\text{victim}}(\pi_{\text{victim}}, \pi_{\text{adversary}})$ in zero-sum games.

# 4 Rational Policy Gradient

Rational policy gradient (RPG) incorporates the constraint from Theorem 3.2 by introducing a new *manipulator* agent policy $\pi_{-i}^M$ for each base agent policy $\pi_i$. Each base agent $i$ now ignores its original objective $O_i(\pi_1, \ldots, \pi_m)$ and instead optimizes to play a best response against its manipulator to enforce the rationality constraint from Equation (1). The manipulator is responsible for the original objectives $O_i$, giving the following two objectives

$$\text{Base agents: } \max_{\pi_i} U(\pi_i, \pi_{-i}^M), \quad \text{Manipulators: } \max_{\pi_{-i}^M} O_i(\pi_1, \ldots, \pi_m). \tag{2}$$

Notice that the manipulators can only influence the original objective indirectly by choosing which policy the base agents should best respond to. Since the rationality constraint has no affect on agents with non-adversarial objectives (e.g., the victim in AT), as an optimization, such base agents keep their original optimization objective: $\max_{\pi_i} O_i(\pi_1, \ldots, \pi_n)$.

The full RPG-modified objectives of several adversarial optimization problems modified are given in Appendix D. In the rest of this section, we give an overview of the principles behind the gradient update for RPG. Further details are available in Section E.

## 4.1 RPG Gradients

RPG approximates the objectives in Equation (2) through a policy gradient update. Let $\theta_i$ and $\theta_{-i}^M$ denote the parameters for policies $\pi_i$ and $\pi_{-i}^M$. The gradient update for the base policies is straightforward

$$\theta_i' \leftarrow \theta_i + \nabla_{\theta_i} U(\theta_i, \theta_{-i}^M). \tag{3}$$

Building atop existing opponent shaping techniques [Foerster et al., 2017], the manipulator takes higher-order gradients through their base agents' update with respect to the adversarial objective

$$\theta_{-i}^M \leftarrow \theta_{-i}^M + \nabla_{\theta_{-i}^M} O_i(\theta_1', \ldots, \theta_m'). \tag{4}$$

Notice that $O_i$ is evaluated at the base agents' parameters $\theta_i'$ after they have taken one gradient step into the future. The manipulators are responsible for the adversarial objective but can only indirectly affect it by altering the base agent's learning process.

## 4.2 Estimating Gradients from Samples

In order to compute gradients in a model-free deep learning setting, we approximate gradients via samples using a surrogate loss that preserve higher-order gradients. Traditional RL surrogate losses are only sufficient for single-order gradients, so we need a different loss for the manipulator gradients that preserves all dependencies (in our case, the manipulator's influence on the base agents' update). We use Loaded DiCE [Farquhar et al., 2019], which is based on DiCE [Foerster et al., 2018] to define a surrogate loss that admits computing unbiased higher-order gradients through autodiff from advantage estimates. Let $E$ be a set of rollouts between different pairs of base agents for the optimization objective $O_i$. The manipulator loss is defined as

$$\mathcal{L}_{\boxdot}^{O_i} = \sum_{e \in E} w_e \sum_t \gamma^t \boxdot(\{a_{j \in B,M}^{t' \leq t}\}) r_e^t, \tag{5}$$

**Algorithm 1** RPG update with lookahead $N$

**input** Policy parameters $\theta_{-i}^M, \theta_i, \forall i \in [m]$.
1: Copy: $\theta_i' \leftarrow \theta_i, \forall i \in [m]$
2: **for** $n$ in $1 \dots N$ **do**
3:     Rollout trajectories under $(\theta_i', \theta_{-i}^M)$
4:     $\theta_i' \leftarrow \theta_i' + \alpha_1 \nabla_{\theta_i'} \mathcal{L}(\theta_i', \theta_{-i}^M)$
5: **end for**
6: Rollout trajectories under $(\theta_1', \dots, \theta_m')$
7: $\theta_{-i}^M \leftarrow \theta_{-i}^M + \alpha_2 \nabla_{\theta_{-i}^M} \mathcal{L}_{\boxdot}^{O_i}(\theta_1', \dots, \theta_m')$
8: $\theta_i \leftarrow \theta_i + \alpha_3 \nabla_{\theta_i} \mathcal{L}(\theta_i, \theta_{-i}^M)$
**output** $\theta_{-i}^M, \theta_i, \forall i \in [m]$.

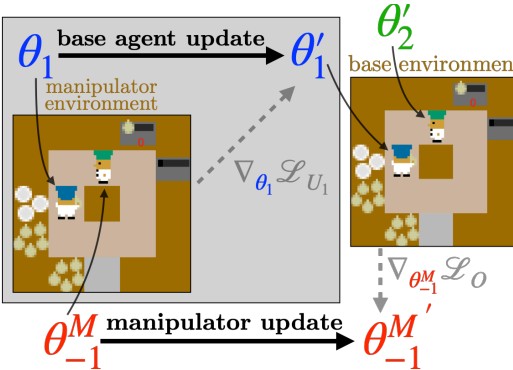

Figure 3: Psuedo code (left) and visualization (right) for the RPG update. The inner box visualizes a base agent's update and the full figure visualizes the corresponding manipulator update taking gradients through the base agent's update.

where $\left\{ a_{j \in B, M}^{t' \leq t} \right\}$ denotes the set of actions from all players when the base agents trains with its manipulator during lookahead, $w_e$ is the weight (e.g., $+1$ for maximize and $-1$ for minimize) specified by the optimization objective, and $r_e^t$ are the associated rewards between the two base agents specified by $e$. $\boxdot$ is the *magic box* operator from [Foerster et al., 2018]. The loss for each base agent is based on a traditional RL surrogate loss $\mathcal{L} = \sum_{e \in E} w_e \sum_t \gamma^t \log \left( \pi(a_t | s_t) \right) r_e^t$, the sum of rewards for different training partners rescaled by the training weight (e.g., $w_e = \epsilon$ for partner-play).

## 4.3 Partner-play Regularization

In deep learning settings, significant distribution shifts between training and evaluation data can lead to poor performance. In our case, base agents are training against manipulators and then evaluated against other base agents: this shift in partner can easily put the base agent's policy out-of-distribution. To minimize the impact of this distribution shift, we introduce a regularization term in the base agent's loss function called *partner-play* regularization: for each base agent $i$, we add to the training data some rollouts of $i$ partnered with each other base agent that $i$ is evaluated against under $O_i$. We scale this by a small value $\epsilon$ so that it only acts as a secondary loss. This prevents $i$'s policy from being out-of-distribution during evaluation and ensures that manipulators cannot optimize adversarial objectives by simply putting the base agents training out-of-distribution.

## 4.4 The Rational Policy Gradient Algorithm

Algorithm 1 shows pseudocode for a single policy update of RPO. Lines 2-5 perform $N$ lookahead steps on the base agent policies by taking gradient steps towards being a best-response to their corresponding manipulator policies (as well as partner-play regularization). Line 6 performs rollouts between all of the updated base agents required to evaluate the original objectives $O_i$ (rollouts between all base agents are shown for simplicity). Line 7 then applies a gradient step on the manipulator policies through the DiCE-modified [Foerster et al., 2018] loss function $\mathcal{L}_{\boxdot}^O$ evaluated on the rollouts from updated base agents. Line 8 simply applies a single gradient step on the base agents (identical to the update on line 4) with respect to a traditional RL surrogate loss. We note that RPG is agnostic to the underlying RL algorithm used to compute the gradients of the loss function.

## 4.5 RPG Algorithms

In this section, we give an overview of the novel algorithms that we introduce using RPG, allowing them to be applied to non-zero-sum games. A more detailed specification along with a visualization of each algorithm is given in Section D as well as a glossary of acronyms and terms in Section A.

**Adversarial policy (AP)-RPG** is used to find flaws in agents. AP-RPG is the RPG variant of adversarial policy (AP) [Gleave et al., 2019]. AP is used in zero-sum settings to find adversarial vulnerabilities in pretrained agents. In both AP and AP-RPG, we refer to the pretrained agent as the *victim*. Meanwhile, the learning agent that attempts to find a vulnerability in the victim is called the *adversary*. In the case of AP-RPG, since we want to prevent the adversary from acting irrationally, it trains against an *adversary manipulator* that guides it towards the adversarial solutions.

**Adversarial training AT-RPG** is used to robustify a learning agent. AT-RPG is the RPG variant of adversarial training (AT) [Gleave et al., 2019]. AT is an extension of adversarial policy that allows the victim to simultaneously learn while the adversary attacks. AT-RPG modifies AP-RPG in the same way by allowing the victim to learn. Since the victim is not subject to any adversarial incentives, it does not require a manipulator.

**PAIRED-RPG** is used to robustify a learning agent using regret. PAIRED-RPG is the RPG variant of protagonist-antagonist induced regret minimization (PAIRED) [Dennis et al., 2020]. PAIRED is an algorithm used for unsupervised environment design and if naively applied to *coplayer design*, the *adversary* can simply learn to sabotage the game when matched with the *protagonist*. PAIRED-RPG fixes this issue by introducing a manipulator for the adversary.

**PAIRED-Attack-RPG** is used to find flaws that maximize an agent's regret. PAIRED-A-RPG is a variant of PAIRED-RPG in which the protagonist is fixed, allowing us to find adversarial policies that the protagonist performs well with, but the victim does not. This finds adversarial examples that do not only minimize score, but maximizes regret.

**Cross-play diversity (XPD)-RPG** is dual purpose: (1) finding a set of meaningfully diverse policies, and (2) generating an auto-curriculum to train robust policies. XPD-RPG is the RPG variant of cross-play diversity (XPD) algorithms underlying approaches such as ADVERSITY [Cui et al., 2023], LIPO [Charakorn et al., 2023], and CoMeDi [Sarkar et al., 2024]. XPD algorithms work by framing diversity as the problem of finding a set of strategies that perform well in self-play but poorly in cross-play, indicating that distinct solutions have been found. However, naively training a population this way leads agents to achieve low cross-play scores by sabotaging the game rather than finding fundamentally incompatible strategies. XPD is often implemented by sequentially training agents, however, we find that simultaneously training all agents with XPD-RPG serves the dual purpose of producing an auto-curriculum that identifies and fixes weaknesses in the population's policies throughout training, providing similar benefits as self-play in zero-sum settings.

## 5 Experiments

Following a description of our experiment setup, results are organized into three subsections according to the following empirical claims, with the last one explored throughout:

**Claim 1** RPG allows the learning of *meaningfully diverse* policies, a set of policies that covers differing strategies within the space.

**Claim 2** RPG algorithms train policies that are more robust to differing partners.

**Claim 3** RPG algorithms find non-trivial adversarial examples in pretrained policies.

**Claim 4** RPG prevents self-sabotage across a range of adversarial algorithms.

Our project webpage includes additional interactive visualizations, demos for playing against our trained agents, and various videos that demonstrate interesting behavior. The glossary in Section A can be useful as a reminder of terms and acronyms throughout.

### 5.1 Experiment Setup

We evaluate our approach in four primary environments: **matrix games** with varying zero-sum, cooperative, and mixed-motive payoffs; several standard **Overcooked** [Carroll et al., 2019] layouts, a modified version of **STORM** [Khan et al., 2023] that requires agents to collect either green or red coins, rewarding them when they collect the same color; and a simplified 2-player version of **Hanabi** [Bard et al., 2020] that contains 3 colors and ranks and a version that contains 4 colors and ranks (instead of the standard 5). See Section F for descriptions and visualization of each of these environments. We use the JaxMARL [Rutherford et al., 2023] implementation for the latter three environments. Section E outlines algorithm implementation and hyperparameter details used.

To evaluate the efficacy of RPG, we compare it against a number of baselines. We include **(1) self-play (SP)** [Samuel, 1959], where policies are learned by training with themselves. We also compare against a selection of existing adversarial learning algorithms. Specifically, we compare against **(2) adversarial policy** [Gleave et al., 2019], **(3) adversarial training** [Gleave et al., 2019], **(4) PAIRED** [Dennis et al., 2020], **(5) a cross-play diversity (XPD)** baseline, and **(6) CoMeDi** [Sarkar et al., 2024], the previous SOTA extension for preventing self-sabotage in cross-play diversity. The XPD baseline we use is very similar to LIPO Charakorn et al. [2023] without a mutual information term and SPWR Cui et al. [2023]. In fully observed settings (e.g., Overcooked), our XPD baseline is

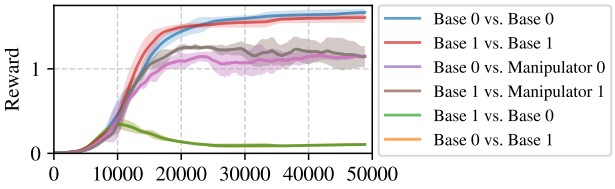

| Algorithm | Self-play | Cross-play |
|-----------|-----------|------------|
| CoMeDi | 220 | 2 |
| XPD | 240 | 1.25 |
| XPD-RPG | 240 | 240 |

Figure 4: **Left:** training curves for each learning agent in XPD-RPG. Self-play rewards increase while cross-play scores decrease (but notably never reach zero). Manipulator's only reach the level of reward necessary to influence the base agents to be diverse. Shaded region shows 95% confidence interval. **Right:** self-play and cross-play rewards for algorithms based on cross-play diversity in the *cramped room* Overcooked layout. Self-sabotage leads to deceptively low cross-play rewards for CoMeDi and XPD while XPD-RPG maintains high score even in cross-play.

effectively identical to ADVERSITY Cui et al. [2023]. See Section C for a detailed discussion of these points.

## 5.2    Claim 1: RPG Finds Meaningfully Diverse Policies

In this section we demonstrate how RPG allows the learning of diverse (i.e, incompatible) policies by solving self-sabotage in cross-play diversity algorithms, whereas existing approaches [Cui et al., 2023, Sarkar et al., 2024] fail.

**STORM.** Figure 4 (left) shows the training curves for XPD-RPG in the original STORM environment. Around episode 10000, the manipulators learn to lead each base agent gather a different colored coin, causing cross-play reward (e.g., base 0 vs. base 1) to drop. The manipulator's rollout rewards eventually plateau since further shaping would do nothing to increase the objective of diversity between base agents. Even in cross-play, agents avoid self-sabotage and continue to receive a positive reward, supporting **Claim 4**. Qualitatively, in cross-play, the agents first each gather an opposite colored coin. Then, once they realize their incompatibility, they adapt their initial strategy, gathering a matching coin in order to receive a partial reward. This highlights that *XPD-RPG only drives down value in cross-play if it is necessary to improve self-play.*

We also test CoMeDi [Sarkar et al., 2024] in the STORM environment and it fails to exhibit any attempt to deviate, quickly driving cross-play reward all the way to zero, even though XPD-RPG shows that this is not necessary for high self-play reward. In order to exemplify this behavior, we modify the original STORM environment so that both agents receive a reward of $-0.1$ on every timestep either agent occupies the top-left square in the grid, giving agents an easy way to sabotage the game if desired. Figure 18 shows the training curves for CoMeDi run in this modified version of the STORM environment. The policy quickly learns to *intentionally* sabotage and receive a large negative reward in cross-play. Meanwhile, XPD-RPG learns to avoid the sabotage state in both self-play and cross-play in this modified environment.

**Overcooked.** Figure 4 (right) shows the self-play and cross-play rewards computed by different algorithms in the *cramped room* Overcooked layout. CoMeDi achieves low cross-play rewards (along with XPD), seemingly a sign of finding incompatible strategies. However, closer inspection reveals that in cross-play, CoMeDi and XPD agents will simply stand in front of the plate dispenser to prevent their partner from finishing any dishes – a clear sign of sabotage. See the project webpage for videos of this behavior. Moreover, high cross-play rewards with *XPD-RPG demonstrate that it is able to avoid sabotage* (**Claim 4**) and that the *cramped room* layout contains very little genuine diversity: agents will only achieve low cross-play reward if they are incentivized to sabotage. This is unsurprising in light of Lauffer et al. [2023], which showed that policies in Overcooked require trivial levels of coordination to be successful.

## 5.3    Claim 2: RPG Trains Robust Policies

In this section, we show that XPD-RPG can learn policies that can generalize across partners. To do so, we train five policies with different seeds for XPD, XPD-RPG, SP (with entropy coefficient 0.01), and SP (with entropy coefficient 0.05). We found that using a large entropy coefficient with SP often encouraged policies to converge on the same strategy (therefore leading to higher robustness within its population). XPD and XPD-RPG both use a population size of two policies.

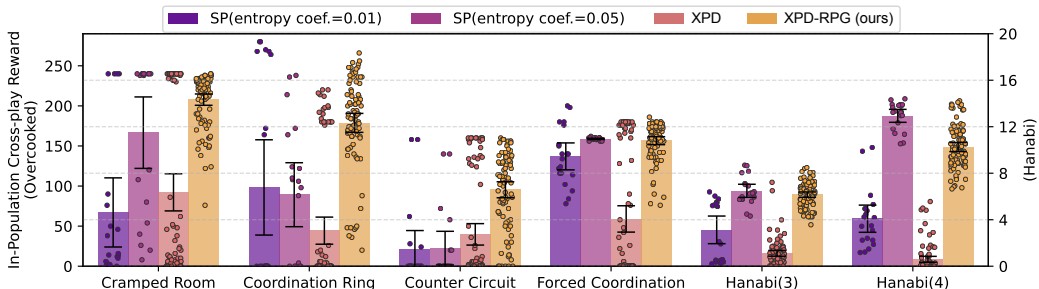

Figure 5: Cross-play grids between different algorithms across environments. Each square represents the average reward from a specific pair of seeds trained by one of the three algorithms when paired as teammates. Standard error < 1 for Overcooked and < 0.1 for Hanabi for all values.

Figure 6: Intra-population cross-play rewards for different algorithms across environments. Points represent pairs of seeds, bar charts represent means, and error bars represent 95% confidence intervals.

Figure 5 shows robustness results for different environments in the form of *cross-play grids*. The project webpage includes an interactive visualization of rollouts from these cross-play grids. In our cross-play grids, each row represents a single seed trained by the specified algorithm. The first five rows represent SP with an entropy coefficient of 0.01 and the second five represent SP with an entropy coefficient of 0.05. The following ten rows represent the five pairs of policies from each run of XPD (since we use a population of size two) and the last ten rows represent XPD-RPG in the same fashion. Each column represents the same so that in diagonal entries, both players come from the same seed. Each square in the cross-play grid represents the average reward of the policies from the two associated seeds played against one another averaged over 1000 rollouts of the game. Figure 6 shows the average *intra-population cross-play* reward for each algorithm: the average of the cross-play scores from within the population of seeds of each individual algorithm.

**Overcooked.** SP policies in *cramped room* and *forced coordination* often tend to converge on a single strategy, especially with high entropy regularization (second set of 5 SP seeds). It's therefore unsurprising that these seeds perform well with one another. However, their adaptability to strategies from other algorithms is very poor – they often achieve close to zero reward with other policies. Take *forced coordination* for example: even the two sets of SP seeds (with different entropy) are completely incompatible. On the other hand, *XPD-RPG performs nearly perfectly with any partner as evident by the right third of the grid achieving high value*. The asymmetry of the grid can be explained by the asymmetry of roles in *forced coordination*: the column player is responsible for picking up onions passed by their partner. XPD-RPG, as the column player, is able to adapt to any strategy for passing the onions while other algorithms are only robust to a single one.

The more complicated layouts of *coordination ring* and *counter circuit* allow multiple solutions and prove a challenge for SP (even with high entropy), achieving very low cross-play rewards. Meanwhile, XPD-RPG is able to maintain much higher cross-play rewards, evident by the higher mean bars in Figure 6 and the fact that it rarely achieves close to zero reward with itself in cross-play.

**Hanabi.** In the 3-color and 4-color version of Hanabi, SP with high entropy achieves significantly higher intra-population rewards than low entropy. We speculate that this is due to the entropy regularization causing policies to converge on very similar strategies. This is somewhat surprising given previous works [Bard et al., 2020, Hu et al., 2020] on Hanabi and might result from our policies lacking history-dependence or because the game is simpler with fewer colors, either of which might make SP policies less likely to develop specialized strategies.

## 5.4   Claim 3: RPG Finds Rational Adversarial Examples

In order to explore the effectiveness of AT-RPG, PAIRED-RPG, AP-RPG, and PAIRED-A-RPG, we use a modified version of STORM, which we call **unobserved STORM**, in which the agents cannot observe their partner's position. This simplifies the strategy space by preventing agents from learning policies that react to their partner (such as how XPD-RPG adapts in cross-play). Table 1 shows the performance of various fixed victims against different types of adversarial attacks in unobserved STORM. The 'Victim' column indicates the algorithm used to train the victim and the 'Training' column the reward that the algorithm achieved by the end of training. The 'AP', 'PAIRED-A-RPG' and 'AP-RPG' columns show the performance against the respective adversarial attacks.

These results demonstrate that (1) *PAIRED-A-RPG and AP-RPG are both able to find weaknesses in fixed policies and neither is susceptible to sabotage*, (2) *PAIRED-RPG and AT-RPG training lead to more robust policies*, indicated by high scores against both PAIRED-A-RPG and AP-RPG attacks. Unsurprisingly, every victim achieves zero reward against the AP attack since the adversary simply learns to self-sabotage the game by collecting no coins. Likewise, the non-RPG variations of algorithms (AT, PAIRED, XPD) fail during training due to self-sabotage. Figures 19 to 21 in Section G.2 show AP, AP-RPG, and PAIRED-RPG attack training curves against these victims.

**Overcooked.** We tested the ability of AP-RPG to discover weaknesses in fixed policies in the *cramped room* Overcooked layout. AP-RPG is able to successfully find weaknesses in a fixed policy trained via self-play, finding an adversarial policy that achieves a reward of 4.6, even though the fixed policy achieves a reward of 240 in self-play. Figure 7 and videos on the project webpage illustrate one of these adversarial examples that AP-RPG finds: instead of simply sabotaging the game like AP, AP-RPG discovers that the victim (blue) assumes that the agents will move around each other clockwise. The manipulator in AP-RPG incentivizes the adversary (red) to instead move in a counterclockwise fashion, a rational strategy, though it happens to be incompatible with the victim.

|            |          | Adversarial Attack |              |        |
|------------|----------|------|--------------|--------|
| Victim     | Training | AP   | PAIRED-A-RPG | AP-RPG |
| PAIRED     | 0.13     | 0.0  | 0.50         | 0.42   |
| PAIRED-RPG | 0.93     | 0.0  | 0.84         | 0.85   |
| AT         | 0.0      | 0.0  | 0.0          | 0.0    |
| AT-RPG     | 0.65     | 0.0  | 0.72         | 0.88   |
| XPD        | 0.00     | 0.0  | 0.00         | 0.00   |
| XPD-RPG    | 0.98     | 0.0  | 0.25         | 0.96   |
| Self-play  | 0.98     | 0.0  | 0.16         | 0.96   |

Table 1: The average reward that "Victims" trained by various algorithms achieve against different adversarial attack types. The "Train" column shows the reward achieved during training and the following columns show the reward against different adversarial attack types. The AP attack trivially achieves zero reward because it suffers from self-sabotage. See Section A for a glossary of acronyms.

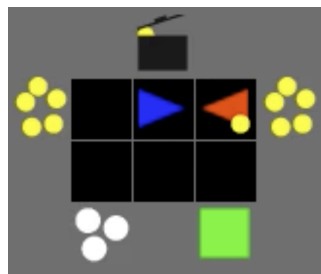

Figure 7: A rational adversarial example found by XPD-RPG. Both agents assume the other will move out of the way, preventing progress.

## 6 Related Works

Prior work [Cui et al., 2023, Sarkar et al., 2024] has identified the phenomenon of self-sabotage in the context of cross-play diversity algorithms and proposed fixes based on making the observation distributions of self-play and cross-play similar. Our experiments show that the approach from Sarkar et al. [2024] fails to prevent sabotage in all of our environments and that [Cui et al., 2023] fails in Overcooked and matrix games, since our XPD baseline is effectively identical to ADVERSITY in fully-observed settings. See Section C for a detailed discussion of why. Several simultaneous works [Ruhdorfer et al., 2025, Wang et al., 2025, Chaudhary et al., 2025] have explored the idea of extending the adversarial training algorithm to the cooperative setting to improve robustness against unseen partners. Specifically, Wang et al. [2025] and Chaudhary et al. [2025] encounter the problem of self-sabotage and attempt to fix it by mixing state distributions across self-play and cross-play and by limiting the adversarial search space via a generative model, respectively. However, none of these methods propose a complete framework for preventing self-sabotage across any adversarial optimization algorithm in the way that we do.

There are several other paradigms for training more performant or robust policies in cooperative settings. Approaches based on the setting of zero-shot coordination Hu et al. [2020], Treutlein et al. [2021], Muglich et al. [2022] aim to learn policies that do not depend on arbitrary symmetries of the game. Several other methods aim make policies robust by training them against a population of agents Vinyals et al. [2019], Rahman et al. [2022] or against human-data-informed policies Carroll et al. [2019], [FAIR], Liang et al. [2024]. Our work differs from all of these by directly applying different forms of adversarial optimization and does not rely on collecting human data.

A significant part of the technical backing of RPG is based on the line of work on *opponent shaping* [Foerster et al., 2017, Kim et al., 2021, Letcher et al., 2018]. To the best of the author's knowledge, RPG represents the first use of opponent shaping in the context of adversarial training. A more recent line of work [Lu et al., 2022, Khan et al., 2023] aims to achieve opponent shaping without having to explicitly compute expensive higher-order gradients. Investigating whether this type of shaping could be incorporated into RPG to improve efficiency would be an exciting line of future research.

## 7 Conclusion

We introduce the problem of *rationality-preserving policy optimization* and give a solution in the form of a gradient-based algorithm called *rational policy gradient* (RPG). We demonstrate the diverse uses of RPG by constructing algorithms that robustify agents, expose rational adversarial examples, and learn genuinely diverse behaviors across general-sum settings. Our results highlight that the success of adversarial optimization in zero-sum training methods can indeed be extended to the general-sum setting and we aim to inspire further research in this direction. Furthermore, we hope that the generality of RPG will enable the development of entirely new classes of adversarial optimization algorithms in general-sum settings.

## 8  Acknowledgment

This research was supported in part by a gift from the Open Philanthropy Foundation to the Center for Human-Compatible AI, by DARPA Contract HR00112490425 (TIAMAT), and the Schmidt AI 2050 Fellowship (Russell). Niklas Lauffer is supported by a National Science Foundation Graduate Research Fellowship and a Cooperative AI Foundation Fellowship.

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

## A    Glossary of terms

| Acronym | Term | Definition |
|---|---|---|
| RPO | Rationality-preserving Policy Optimization | Section 3.2, Equation (1) |
| RPG | Rational Policy Gradient | Section 4, Algorithm 1 |
| AP-RPG | Adversarial Policy – RPG | Section 4.5 Section D.2 |
| AT-RPG | Adversarial Training – RPG | Section 4.5 Section D.3 |
| PAIRED-RPG | Protagonist–Antagonist Induced Regret Min. – RPG | Section 4.5 Section D.4 |
| PAIRED-A-RPG | PAIRED-Attack – RPG | Section 4.5 Section D.5 |
| XPD-RPG | Adversarial Diversity – RPG | Section 4.5 Section D.6 |
| AP | Adversarial Policy | [Gleave et al., 2019] |
| AT | Adversarial Training | [Gleave et al., 2019] |
| PAIRED | Protagonist–Antagonist Induced Regret Minimization | [Dennis et al., 2020] |
| XPD | Adversarial Diversity | [Cui et al., 2023, Charakorn et al., 2023] |
| SP | Self-play | [Samuel, 1959] |
| CoMeDi | Cross-play optimized, Mixed-play enforced Diversity | [Sarkar et al., 2024] |

Table 2: Glossary of terms and acronyms.

## B    Limitations

RPG relies on higher-order gradients through agents' learning updates, which introduces computational overhead, especially in high-dimensional domains. However, this overhead is roughly linear in the number of optimizing agents in the problem and is indepedent of any other aspects of the problem. Empirically, we see that cross-play diversity (XPD) is roughly two times slower (in wall-clock time) throughout our experiments since it doubles the number of agents and XPD-RPG is an extra three times slower than that through the introduction of a manipulator agent for each base agent. Otherwise, RPG algorithms experience only a constant overhead as the size of the problem grows. Estimating higher-order gradients from samples has high variance, so large batch sizes are required to stabilize learning in our experiments although recent works [Engstrom et al., 2025] have significantly improved the stability of meta-gradients and could be extended to our work.

While RPG enforces rationality constraints by ensuring that base agents only train to maximize their utility, we currently have no formal guarantee of convergence to truly rational strategies. However, it would be useful for future work to explore under which conditions RPG ensures a solution to RPO.

## C    Comparison to existing attempts to solve self-sabotage

In this section, we theorize why existing approaches to preventing self-sabotage fail. In order to match the setting of existing work on self-sabotage [Cui et al., 2023, Sarkar et al., 2024], we investigate the setting of the *cross-play diversity* (XPD) algorithms. XPD algorithms aims to generate a diverse population of policies by maximizing self-play scores and minimizing cross-play scores within the population. Our XPD baseline is very similar to LIPO Charakorn et al. [2023] except we use an average instead of a max to aggregate cross-play scores and do not add any additional mutual information terms to the loss. Our XPD baseline is nearly identical to ADVERSITY [Cui et al., 2023] in fully-observed settings since there is no partially-observed history to do belief reinterpretation over. The only remaining distinction is that our XPD baseline rolls out SP and XP in distinct rollouts instead of mixing within the same rollout as ADVERSITY does.

Now we turn to observing why traditional approaches to solving self-sabotage in XPD algorithms have failed. Consider running XPD with a population of size two in the following matrix game:

|  |  | Player Y | | | |
|---|---|---|---|---|---|
|  |  | $C$ | $D$ | $E$ | $F$ |
| Player X | $A$ | 1 | 0.9 | −1 | 0 |
|  | $B$ | 0 | −1 | 0.9 | 1 |

Figure 8: The payoff matrix for a cooperative game.

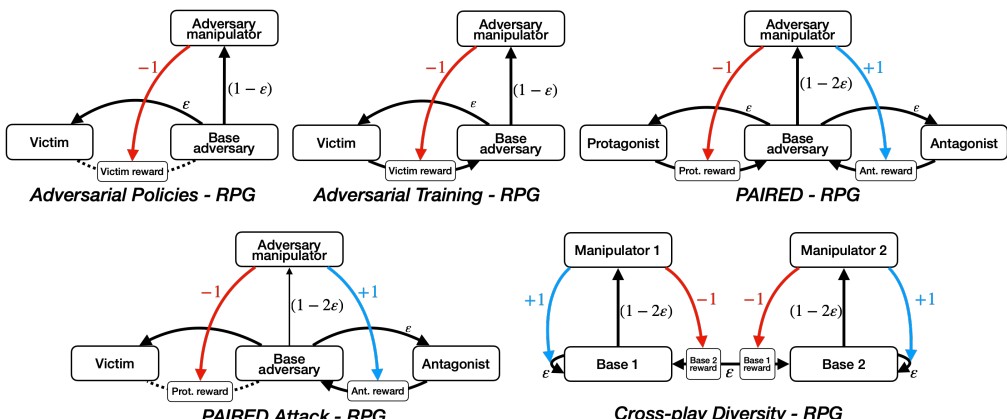

Figure 9: Graphical representation of the RPG algorithms described in section D.1. Nodes represent optimizing agents and outgoing edges represent their optimization objectives. Weights on edges scale the importance of different objectives such that negative weights indicate minimization.

Running the original XPD algorithm (along with those from [Cui et al., 2023, Sarkar et al., 2024]) with a population of size two will lead to the solution $\{(A, D), (B, E)\}$, achieving self-play scores of $0.9$ and cross-play scores of $-1$. Both policies maximize the objectives of their respective algorithm by irrationally playing $D$ and $E$ instead of $C$ and $F$, respectively. They sacrifice a little bit of score in self-play to greatly decrease the cross-play score. However, notice that the actions $D$ and $E$ are strictly worse no matter what the partner plays – clearly an irrational choice.

As identified in [Cui et al., 2023, Sarkar et al., 2024], true diversity should only come from a decreases in cross-play scores that are *necessary* to improve self-play scores within the population. Choosing action $D$ over $C$ is the opposite: sacrificing self-play score to artificially reduce cross-play score. XPD-RPG finds the desired solution $\{(A, C), (B, F)\}$ because it constrains the diversity to rational policies and only allows a decrease in cross-play if it benefits self-play However, just like ordinary XPD, the algorithms presented in [Cui et al., 2023, Sarkar et al., 2024] find the undesired solution $\{(A, D), (B, E)\}$. Both of the algorithms presented in these works are based on the idea of confusing agents during training so that they do not know whether they are currently training in self-play or cross-play. They do this by altering RL training so that the distributions of observations seen during self-play and cross-play are as similar as possible. However, the example in Figure 8 demonstrates that self-sabotage can happen, even if there's only a single observation in the game. No amount of confusion about observation distributions can prevent self-sabotage, so the approaches in [Cui et al., 2023, Sarkar et al., 2024] fail in situations like this. The experiments shown in Figure 4 and Figure 18 demonstrate that these approaches fail in STORM and Overcooked, too, indicating that the structure of the game in Figure 8 may be present in more complicated environments.

# D   RPO objectives

## D.1   A Graphical Language for RPG Algorithms

In order to simplify specifying and interpreting RPG algorithms, we introduce a graphical language. Agents (base agents and manipulators) are represented by nodes and an agent's loss function is represented by a collection of triples that can be visually represented by outgoing edges in the graph. A triple is defined by the optimizing agent (i.e. the node that the edge is coming out of) and the two agents being evaluated.

Figure 9 shows the graph for all RPG algorithms. Let us use AP-RPG as an example. It includes three agents: the victim, adversary base agent, and adversary manipulator. The victim is fixed, indicated by having no solid outgoing edges. The adversary base agent has two outgoing edges, one for training against its manipulator (with weight $1 - \epsilon$), and one for partner-play against the victim base agents (with weight $\epsilon$). The adversary manipulator has a single outgoing edge pointing to the victim's reward when the adversary base agents and victim base agents are evaluated against one another. The weight of -1 on this edge indicates that the adversary manipulator's loss wants to minimize this value.

### D.2 Adversarial Policy-RPG

The objective for adversarial policy-RPG is

$$\min_{\pi_{\text{adversary}}^M} U_{\text{victim}}(\pi_{\text{adversary}}, \pi_{\text{victim}}),$$
$$\text{s.t.} \quad \pi_{\text{adversary}} \in BR(\pi_{\text{adversary}}^M). \tag{6}$$

Notice that AP-RPG is identical to AT-RPG except that the the victim is fixed.

### D.3 Adversarial Training-RPG

The objective for adversarial training-RPG is

$$\min_{\pi_{\text{adversary}}^M} U_{\text{victim}}(\pi_{\text{adversary}}, \pi_{\text{victim}}),$$
$$\max_{\pi_{\text{victim}}} U_{\text{victim}}(\pi_{\text{adversary}}, \pi_{\text{victim}}),$$
$$\text{s.t.} \quad \pi_{\text{adversary}} \in BR(\pi_{\text{adversary}}^M). \tag{7}$$

### D.4 PAIRED-RPG

The objective for PAIRED-RPG is

$$\max_{\pi_{\text{adversary}}^M} U_{\text{victim}}(\pi_{\text{adversary}}, \pi_{\text{antagonist}}) - U_{\text{victim}}(\pi_{\text{adversary}}, \pi_{\text{protagonist}}),$$
$$\max_{\pi_{\text{protagonist}}} U_{\text{victim}}(\pi_{\text{adversary}}, \pi_{\text{protagonist}}),$$
$$\max_{\pi_{\text{antagonist}}} U_{\text{victim}}(\pi_{\text{adversary}}, \pi_{\text{antagonist}}),$$
$$\text{s.t.} \quad \pi_{\text{adversary}} \in BR(\pi_{\text{adversary}}^M). \tag{8}$$

### D.5 PAIRED-Attack-RPG

The objective for PAIRED-A-RPG is

$$\max_{\pi_{\text{adversary}}^M} U_{\text{victim}}(\pi_{\text{adversary}}, \pi_{\text{antagonist}}) - U_{\text{victim}}(\pi_{\text{adversary}}, \pi_{\text{protagonist}}),$$
$$\max_{\pi_{\text{antagonist}}} U_{\text{victim}}(\pi_{\text{adversary}}, \pi_{\text{antagonist}}),$$
$$\text{s.t.} \quad \pi_{\text{adversary}} \in BR(\pi_{\text{adversary}}^M). \tag{9}$$

Notice that PAIRED-A-RPG is identical to PAIRED-RPG except that the protagonist (which acts as the victim) is fixed.

### D.6 Adversarial Diversity-RPG

The objective for cross-play diversity-RPG with a population size of $m$ is

$$\max_{\pi_1^M \dots, \pi_m^M} \sum_{i \in [m]} \left( U_i(\pi_i, \pi_i) - \frac{\lambda}{m-1} \sum_{j \in [m], j \neq i} U_i(\pi_i, \pi^j) \right) \tag{10}$$
$$\text{s.t.} \quad \pi_i \in BR(\pi_i^M) \quad \forall i \in [m],$$

where $\lambda$ weighs the relative importance of minimizing cross-play. Our experiments use $\lambda = 0.25$. Note that we only apply XPD-RPG to cooperative settings.

# E   RPG Details

## E.1   Implementation details

**RL algorithm.** Our implementation of RPG uses actor-critic with a single epoch and update as the RL algorithm. We initially considered trying PPO, but realized that clipping might interfere with the higher-order gradients, leaving the implementation of more sophisticated RL algorithms to future work. For the same reason, only the manipulators use a max gradient norm. We use a separately learned centralized critic for each pair of agents rolled out in the underlying game. All of our experiments also use entropy regularization.

**Choice of optimizer.** Since the manipulators need to take gradient's through the base agent's update, the base agent's optimizer update must be differentiable. Many modern optimizers for deep learning incorporate methods such as *momentum*, which might make it more difficult for the manipulator's to estimate their influence on their base agent. In our experiments, we show that our methods work whether we use vanilla SGD or adam optimizers [Kingma, 2014].

**Normalization.** We also find that properly normalizing advantage estimates for each agent is important: each agent normalizes the advantage estimates associated with each of their training partners together. We also tried normalizing advantages between training partner's independently but learning proved to be more unstable.

**Variable learning rates.** Using different learning rates between the manipulator and the base agents is often important. We find that it is often necessary to use a significantly larger manipulator learning rate than base agent learning rate to magnify the manipulator's ability to influence the base agent's learning. We also find that using different learning rate in the base agent lookahead vs. its actual update step is important. A larger inner base agent learning rate allows the manipulator to project the base agent's update further into the future, similar to using a additional lookahead step, except without the added computational burden.

**Partner-play.** Partner-play regularization also makes a manipulator's optimization problem more difficult: if a manipulator wants to minimize the score between two base agents, it cannot simply make sure its base agent's training is trivially out-of-distribution. We find that in the case of XPD-RPG, a small partner-play (0.05 - 0.1) is more effective at finding genuinely diverse solutions while a large partner-play (0.1 - 0.2) is more effective at robustifying policies. Too large of a partner-play eliminates the manipulator's influence over the base agent entirely, rendering RPG useless.

## E.2   RPG hyperparameters

Table 3 contains the RPG hyperparemeters used in the experiments for each of our environments.

Table 3: RPG hyperparameters for different environments

| Hyperparameter | Matrix | STORM | Overcooked | Hanabi |
| --- | --- | --- | --- | --- |
| Architecture | 64x64 | 64x64 | 64x64 | 512x512 |
| Optimizer | SGD | Adam | Adam | Adam |
| Manipulator LR | $1 \times 10^{-2}$ | $8 \times 10^{-3}$ | $1 \times 10^{-3}$ | $2.5 \times 10^{-3}$ |
| Base LR | $1 \times 10^{-2}$ | $1 \times 10^{-4}$ | $2 \times 10^{-4}$ | $5 \times 10^{-3}$ |
| Base lookahead LR | $1 \times 10^{-1}$ | $4 \times 10^{-4}$ | $2 \times 10^{-4}$ | $5 \times 10^{-3}$ |
| Batch size | 128 | 256 | 512 | 128 |
| Discount factor ($\gamma$) | 0.95 | 0.99 | 0.99 | 0.99 |
| GAE parameter ($\lambda$) | 0.95 | 0.99 | 0.99 | 0.99 |
| Loaded DiCE coef ($\lambda$) | 0.95 | 0.99 | 0.99 | 0.99 |
| Value function coef. | 0.5 | 0.5 | 0.5 | 0.5 |
| Entropy coef. | 0.0 | 0.01 | 0.01, SP: 0.05 | 0.01, SP: 0.05 |
| Manipulator max gradient norm | 0.5 | 0.5 | 0.5 | 0.5 |
| Partnerplay coefficient | 0 | 0.15 | 0.1 | 0.1 |

### E.3 Compute Resources

Experiments were all performed on single GPUs (a mix of A4000s and A6000s) on a local SLURM cluster using 32 cpu cores and 50Gb of RAM. Seeds for all experiments took between a minute up to 24 hours to run.

# F  Environment descriptions

## F.1  STORM

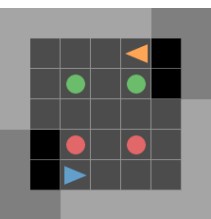

Figure 10: A visualization of the storm environment with green and red coins and the agents in orange and blue. Their 5x5 observation windows are illuminated in front of them.

The Spatial-Temporal Representations of Matrix Games (STORM) [Khan et al., 2023] environment is visualized in Section F. The agents move around the grid and collect green and red coins while only observing a limited area directly in front of them. The agents can 'interact' by getting near each other and playing the 'interact' action. When the agents interact, a coin is randomly chosen from the ones they have picked up. The agents both get a reward of +1 if they interact with the same colored coin, and 0 otherwise.

Our experiments also investigate two modifications of the original STORM environment. In the first modification, which we refer to as *unobserved STORM*, agents' observations no longer include their partner's position. We use this modification to simplify analysis by eliminating the possibility of strategies that react to their partner's policy. In the second modification, both agents receive a reward of -0.1 when either agent stands in grid position (0,0). We use this modification to exemplify sabotaging behavior exhibited by the CoMeDi [Sarkar et al., 2024] baseline.

## F.2  Overcooked

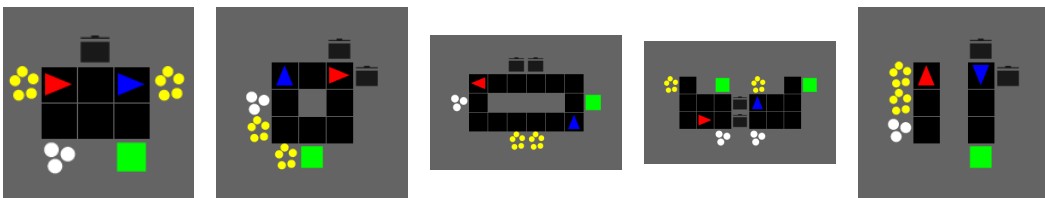

Figure 11: Visualization of various Overcooked layouts in their initial positions. From left to right: Cramped Room, Coordination Ring, Counter Circuit, Asymmetric Advantages, and Forced Coordination.

Overcooked [Carroll et al., 2019] is a cooperative game where players control chefs working together in a kitchen to prepare and serve meals under time constraints. The agents need to work together to gather multiple ingredients into a pot, wait for the soup to cook, and then plate and deliver the dish. We perform experiments in a variety of Overcooked layouts.

## F.3  Hanabi

Hanabi [Bard et al., 2020] is a cooperative, imperfect-information card game that has been a cornerstone of recent coordination benchmarks. Players work together to arrange a shared set of colored, numbered cards in order. Each player holds their cards facing outward, unable to see their own hand but able to see others'. Players provide limited hints to teammates about their hands, constrained by a finite set of clue tokens. The challenge lies in deducing one's own cards and playing them correctly without direct knowledge. It is difficult to coordinate in Hanabi because strategies often rely on arbitrary signaling strategies to communicate information to teammates. Hanabi is typically played with 5 colors and 5 ranks, but we experiment with a version that uses 3 colors and 3 ranks as well as a version that uses 4 colors and 4 ranks.

# G Additional results

## G.1 Matrix Games

### G.1.1 RPG stabilizes learning dynamics.

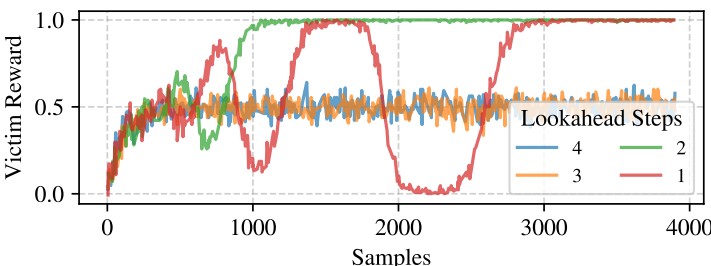

Figure 12: A larger RPG lookahead stabilizes learning dynamics and allows them to converge on an unstable equilibrium.

It is well known that multi-agent problems can have unstable learning dynamics [Claus and Boutilier, 1998, Bowling and Veloso, 2002]. However, a large enough number of lookahead steps in RPG helps stabilize learning, leading to convergence. Figure 12 shows the reward of the learned victim policy in various runs of AT-RPG for the matrix game from Figure 2. The solutions converge on playing an equal mixture between actions $A$ and $B$ even though this is an unstable equilibrium. By allowing the manipulator to look several steps forward through the learning of the base agents, it can anticipate that the victim could respond to a deviation which would ultimately decrease the manipulator's objective. We observed the effect in RPG algorithms across a variety of matrix games. See Section G.1 for more details.

### G.1.2 Understanding how RPG behaves through various matrix games

|  |  | Player $Y$ | | |
|---|---|---|---|---|
|  |  | $C$ | $D$ | $E$ |
| Player $X$ | $A$ | 1 | 0.9 | 0 |
|  | $B$ | 0 | $-1$ | 1 |

Figure 13: A common-payoff game with a dominated action.

Consider the problem of finding two different pairs of diverse strategies in the matrix game in Figure 13 using cross-play diversity (XPD). Imagine for a moment that column $D$ did not exist. Clearly, the most diverse population of successful strategy pairs would then be the solutions $(A, C)$ and $(B, E)$: both pairs of strategies get the maximum self-play score of 1 and the minimum cross-play score of 0. Now, consider adding column $D$ back in: Player $Y$ could change their action in the first pair from $C$ to $D$. This greatly improves the diversity metric by decreasing the cross-play score to -1 while only losing a little bit of score in self-play. However, Player $Y$ choosing to play action $D$ over $C$ is clearly a case of sabotage: not only are they intentionally reducing their score in cross-play, they are doing it at the cost of self-play performance! In fact, action $D$ is *strictly dominated* by action $C$: it performs worse in all scenarios. Not only do agents artificially sabotage their performance in cross-play by playing action $D$, they do it at the cost of self-play performance. Existing work [Cui et al., 2023, Sarkar et al., 2024] fails to prevent sabotage in these types of scenarios because it does not depend on a different in observation distributions between cross-play and self-play.

Figure 14 shows a mixed-motive game with a sabotage option. AT and PAIRED sabotage by playing action $E$. AT-RPG converges on playing a uniform mixture across the two rational policies while PAIRED-RPG converges on one of the two equilibria. XPD converges on one sabotage policy and one at equilibrium while XPD-RPG converges on the two equilibrium.

Figure 15 shows a cooperative game. Suppose the victim in AT is Player $X$ and the adversary is Player $Y$. In this case, AT oscillates between playing $C$ and $D$ to minimize the victim's reward. On

Player $Y$

|  | | $C$ | $D$ | $E$ |
|---|---|---|---|---|
| Player $X$ | $A$ | $3, 2$ | $0, 0$ | $-1, -1$ |
| | $B$ | $1, 1$ | $2, 3$ | $-1, -1$ |

Figure 14: A mixed-motive game with a sabotage option.

Player $Y$

|  | | $C$ | $D$ | $E$ |
|---|---|---|---|---|
| Player $X$ | $A$ | $0.9$ | $0$ | $1$ |
| | $B$ | $0$ | $0.9$ | $1$ |

Figure 15: A cooperative game.

the other hand, since such an adversary is irrational, the adversary in AT-RPG will just play action $E$. Likewise, XPD finds artificially diverse solutions by playing $(A, C)$ and $(B, D)$. Both sets of policies in XPD-RPG just play action $E$, the only rational action.

Player $Y$

|  | | $C$ | $D$ |
|---|---|---|---|
| Player $X$ | $A$ | $0, 0$ | $-1, 1$ |
| | $B$ | $1, -1$ | $-10, -10$ |

Figure 16: A mixed-motive game of 'chicken'.

Figure 16 shows the mixed-motive game of 'chicken'. In this case, AT and AT-RPG solutions align and always play action $D$ to minimize the victim's reward, forcing them to play the conservative action of $A$.

Player $Y$

|  | | $C$ | $D$ | $E$ |
|---|---|---|---|---|
| | $A$ | $0, 0$ | $-1, 1$ | $1, -1$ |
| Player $X$ | $B$ | $1, -1$ | $0, 0$ | $-1, 1$ |
| | $C$ | $-1, 1$ | $1, -1$ | $0, 0$ |

Figure 17: A zero-sum game of 'rock-paper-scissors'.

Figure 17 shows the zero-sum game of 'rock-paper-scissors'. In this case, AT and AT-RPG solutions have very similar behavior in which the adversary and victim oscillate between picking rock, paper, or scissors during training (similar to the oscillating behavior in 12). We found that increasing the lookahead (e.g. greater than 8) in AT-RPG led to this behavior stabilizing and converging on playing the strategy that randomizes across the three actions (in the same way lookahead stabilizes the oscillations in 12).

## G.2 STORM

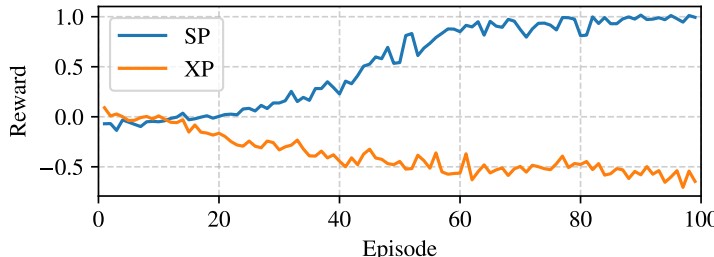

Figure 18: CoMeDi learns to sabotage in a modified version of STORM that allows easy sabotage.

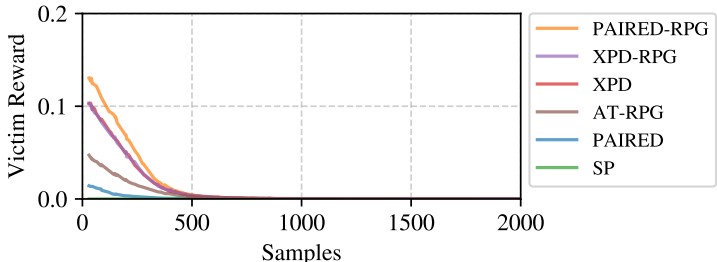

Figure 19: Training curves for AP applied to various victims in the unobserved STORM environment. As expected, AP immediately learns to sabotage, revealing nothing useful about the robustness of each victim.

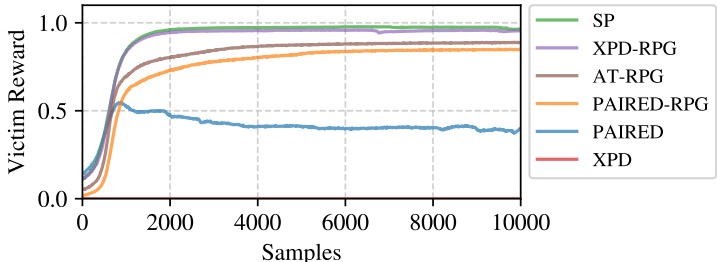

Figure 20: Training curves for AP-RPG applied to various victims in the unobserved STORM environment. AP-RPG avoids the problem of sabotage but inconsistently finds weaknesses in victim policies.

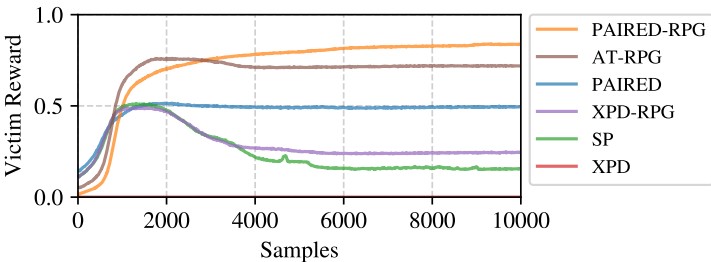

Figure 21: Training curves for PAIRED-RPG attack applied to various victims in the unobserved STORM environment. PAIRED-RPG attack consistently finds weaknesses in victim policies while avoiding the problem of sabotage. These results also highlight that PAIRED-RPG and AT-RPG can produce more adversarially robust policies.

