# OpenReview forum: "Robust and Diverse Multi-Agent Learning via Rational Policy Gradient"
_NeurIPS.cc/2025/Conference — NeurIPS 2025 poster_

### Official Review · Reviewer_6nz8 · 2025-06-25

**Clarity:** 3
**Significance:** 3
**Originality:** 3
**Rating:** 4
**Confidence:** 3

**Summary:**

In this paper, a method called RPO is proposed which is a formalism for adversarial optimization. The proposed method can avoid self-sabotage by ensuring agents remain rational. In detail, authors use opponent shaping techniques to optimize the adversarial objective. In summary， the contribution of this paper follows:
(1) A novel algorithm, PRO, is proposed which can avoid self-sabotage by ensuring agents remain rational.
(2) Authors empirically demonstrate the proposed algorithm and the proposed methos outperform existing baselines in popular cooperative environments.

**Questions:**

In this paper, a formalism called RPO is proposed which can overcomes the issue of self-sabotage. However, there are some problems:
 (1) The experiments are too toy. Four primary environment such as matrix games, Overcooked, STORM,Hanabi are performed. These experiments are too easy and only few agents play. More complex experiments should be performed,such as Markov Games with more agents, or "StarCraft II"
(2) The details of neural networks for policy network should be given and study that how neural network parameters affect the results.
(3)  "Conclusion" should be added at the last section of this paper.
(4) Only pseudocode is available. How to obtain the open-source code and reproduce the experimental results?

**Ethical Concerns:**

["NO or VERY MINOR ethics concerns only"]

**Final Justification:**

More complex experiments should be performed to validate your proposed method. For example more players or large-scale games. There are no novel results, thereby, I will keep my score.

**Limitations:**

Yes.

**Paper Formatting Concerns:**

There is no "conclusion" in section 6.

**Quality:**

3

**Strengths And Weaknesses:**

The strengths of this paper are:
(1) The paper introduces a formalism called RPO that overcomes the issue of self-sabotage in any adversarial optimization algorithm.
(2) The paper introduce a gradient-based deep learning algorithm called RPG.
(3) The paper empirically demonstrates the method outperforms existing baselines.
The weaknesses of this paper are:
(1) The experiments are too toy. Four primary environment such as matrix games, Overcooked, STORM,Hanabi are performed. These experiments are easy and only few agents.
(2) Is it suitable for large number agents, or large scale Markov Game such as "Go" or "MultiAgent System",or StarCraft II?
(3) There are some format error such as lacking "conclusion" at the last section of this paper.

---

> ### Author Rebuttal · Authors · 2025-07-31
>
> Thank you for your review. We are grateful that you highlighted our “novel algorithm” and that the paper "empirically demonstrates the method outperforms existing baselines". We address each point individually:
>
> > “(1) The experiments are too toy. Four primary environment such as matrix games, Overcooked, STORM, Hanabi are performed. These experiments are too easy and only few agents play. More complex experiments should be performed,such as Markov Games with more agents, or "StarCraft II"
>
> Our experiments use the same environments as many recent related research. Overcooked and Hanabi are regularly used to test various SOTA algorithms for cooperative environments. Indeed, Hanabi was identified as a challenge domain (Bard, Nolan, et al. "The hanabi challenge: A new frontier for AI research.") that remains unsolved and in which existing algorithms (i.e., the algorithms used to solve Go) fail. Furthermore, the game is far from small considering how many possible combinations and sequences of cards each player can have and the fact that players need to communicate hidden information.
>
> > “Is it suitable for large number agents, or large scale Markov Game … ?”
>
> In principle, all of our algorithms should scale to a large number of agents and larger Markov games. We observed a constant linear overhead (between 1.5-3 times) over existing algorithms across our experiments (a detail that we will include in the final version of the paper). We believe that further implementation and scaling compute resources will allow the algorithms to scale to these larger settings with the same benefits observed in our experiments.
>
> > “There are some format error such as lacking ‘conclusion’ …”
>
>  We will add a proper conclusion with the extra page allowed for the camera-ready version. Here is a draft:
>
> We introduce the problem of *rationality-preserving policy optimization* and give a solution in the form of a gradient-based algorithm called *rational policy gradient (RPG)*. We demonstrate the diverse uses of RPG by constructing algorithms that robustify agents, expose rational adversarial examples, and learn genuinely diverse behaviors across general-sum settings. Our results highlight that the success of adversarial optimization in zero-sum training methods can be extended to the general-sum setting and we aim to inspire further research in this direction. Furthermore, we hope that the generality of RPG will enable the development of entirely new classes of adversarial optimization algorithms in general-sum settings.
>
> > “Details of neural networks should be given...”
>
> Thank you for catching this. Most hyperparameters for the experiments are given in appendix E.3 but we missed details about the neural network architecture: we used a network with two hidden layers of size 64 in Overcooked, STORM, and the matrix games, and two hidden layers with size 512 in Hanabi. The architecture is taken from a popular open-source implementation for self-play policies in these environments (Rutherford, Alexander, et al. "Jaxmarl: Multi-agent rl environments and algorithms in jax."). These details will be added in the final version.
>
> > “‘Only pseudocode is available. How to obtain the open‑source code?’”
>
> We will include a link to a full open-source implementation of the code in the final version of the paper along with scripts for reproducing the paper’s experiments.

---

> > ### Comment · Reviewer_6nz8 · 2025-08-03
> >
> > Thank you for your  detailed response. However, some of my concerns remain unaddressed. More complex experiments should be performed to validate your proposed method. For example more players or large-scale games. There are no novel results, thereby, I will keep my score.

---

### Official Review · Reviewer_gj2Y · 2025-07-03

**Clarity:** 2
**Significance:** 3
**Originality:** 3
**Rating:** 4
**Confidence:** 3

**Summary:**

This paper introduces Rationality-preserving Policy Optimization (RPO), a new framework for adversarial optimization in multi-agent reinforcement learning that avoids the common failure mode of self-sabotage in cooperative or general-sum environments. Traditional adversarial training methods, successful in zero-sum games, perform poorly in cooperative settings because they incentivize agents to irrationally minimize each other's rewards, harming collective outcomes. To address this, the authors propose Rational Policy Gradient (RPG), a learning algorithm that trains agents to remain rational—optimal with respect to some teammate policy—while optimizing adversarial objectives through a novel “manipulator” agent structure. RPG enables robust, diverse, and self-consistent policy learning without explicit sabotage. Authors argue that RPG-based methods outperform existing baselines in robustness, diversity, and rationality across several empirical benchmark environments.

**Questions:**

1. Could the authors clarify what the variable n represents in Equation (4) and Algorithm 1?
2. Are there any guidelines or intuitions for selecting the hyperparameters $\alpha_1, \alpha_2, \alpha_3$ in Algorithm 1? For example, which one should be larger?
3. There seems to be an inconsistency between the text and figure caption: Line 260 states that five policies are trained with different random seeds, while the caption of Figure 5 mentions two seeds. Please clarify the exact number of seeds used when evaluating robustness claims.
4. In Figure 7, could you confirm whether the values under the adversarial attack columns refer to:
(a) performance during adversarial training, or (b) performance after training without adversaries and testing with adversarial attacks? Additionally, as the content is tabular, it may be clearer to present it as a table rather than a figure.

**Ethical Concerns:**

["NO or VERY MINOR ethics concerns only"]

**Final Justification:**

Most of my questions are for clarification and explanation. I understand more about this work after rebuttal, so I will keep my final score as shown.

**Limitations:**

yes

**Quality:**

2

**Strengths And Weaknesses:**

## Strengths
1.  Strengthen the contribution to explicitly connect the proposed unified framework to existing adversarial optimization algorithms and clarify how different methods can be seen as special cases.
2. The toy example is a clear and well-constructed illustration of the problem, and the insights drawn from it are compelling.
3. The paper gives an insightful theoretical framing that distinguishes rationality constraints from conventional adversarial training.

## Weaknesses
1. Regarding Claim 3: Finding rational adversarial examples, I am not fully convinced. For example, Line 306 states that PAIRED-A-RPG is more effective than AP-RPG at identifying policy weaknesses—but it is unclear how this conclusion is drawn. Could this be due to the added noise acting as an overly strong perturbation? Instead, I suggest evaluating each policy using a learned "worst-case but rational" adversary, and then reporting results under both (a) this learned attack and (b) a no-attack baseline. This would offer a more rigorous assessment of policy robustness.

2. I understand the constraints imposed by the page limit, but including a conclusion section is important for completeness and helps reinforce the key takeaways of the paper.

---

> ### Author Rebuttal · Authors · 2025-07-31
>
> Thank you for your review. We appreciate that “the paper gives an insightful theoretical framing” and the “insights drawn from [the examples] are compelling”. We address each point individually:
>
> > “Regarding Claim 3: Finding rational adversarial examples… it is unclear how this conclusion is drawn.”
>
> Figure 7 is quite nuanced and we will clarify each of the columns in the final version. If we understand your questions correctly, then your suggested (a) is in fact the AP-RPG and PAIRED-A-RPG columns: these are two different methods for constructing “worst-case but rational” adversaries against the different victims represented by the rows. The conclusion “PAIRED-A-RPG is more effective than AP-RPG” is drawn because between those two columns, PAIRED-A-RPG more consistently finds the worst-case (by achieving the lower score) than AP-RPG. Your suggestion for “(b) a no-attack baseline” is represented by the column “Training”: this shows the performance that the policy gets at the end of training (when paired with itself). Finally, the column “AP” simply demonstrates that the AP algorithm acts irrationally and always achieves zero reward. Thank you for raising these points since we believe this clarification will help the reader better understand the evidence supporting claim 3.
>
> > “Include a conclusion section …”
>
>  We will add a proper conclusion with the extra page allowed for the camera-ready version. Here is a draft:
>
> We introduce the problem of *rationality-preserving policy optimization* and give a solution in the form of a gradient-based algorithm called *rational policy gradient (RPG)*. We demonstrate the diverse uses of RPG by constructing algorithms that robustify agents, expose rational adversarial examples, and learn genuinely diverse behaviors across general-sum settings. Our results highlight that the success of adversarial optimization in zero-sum training methods can be extended to the general-sum setting and we aim to inspire further research in this direction. Furthermore, we hope that the generality of RPG will enable the development of entirely new classes of adversarial optimization algorithms in general-sum settings.
>
> > “Could the authors clarify what the variable n represents … ?”
>
> The n in equation (4) should be an m, the number of policies in the optimization problem. The capital N variable in Algorithm 1 is the number of lookahead steps. We will add this clarification the caption of Figure 3. Thank you for improving the clarity of the paper with these comments.
>
> > “Are there any guidelines or intuitions for selecting the hyperparameters in Algorithm 1?”
>
> This is a great question, a discussion of these hyperparameters can be found in appendix E.2 under section “Variable learning rates”, but will consider elevating it to the main text to make this point more visible. We often found it important to use a larger manipulator learning rate (alpha_2) than base agent learning rate (alpha_3) to magnify the manipulator’s ability to influence the base agent’s learning. A larger inner base agent learning rate (alpha_1) allows the manipulator to project the base agent’s update further into the future, similar to using additional lookahead steps.
>
> > “There seems to be an inconsistency between the text and figure caption: Line 260 states that five policies are trained with different random seeds, while the caption of Figure 5 mentions two seeds. Please clarify the exact number of seeds used when evaluating robustness claims.”
>
> We agree that the wording is confusing and will be fixed in the final version. There are indeed 5 seeds per algorithm. The cross-play grids in Figure 5 depict the average score that a specific pair (i.e., “two seeds”) of policies get when paired as teammates in the two-player game.
>
> > “In Figure 7, could you confirm whether the values under the adversarial attack columns refer to: (a) performance during adversarial training, or (b) performance after training without adversaries and testing with adversarial attacks?
>
> It is (b), the performance when testing with an adversarial attack (although some of the rows were also trained with adversarial methods). We agree that the explanation of this figure needs to be improved and will clarify this point along with your previous comments related to this figure.
>
> > Additionally, as the content is tabular, it may be clearer to present it as a table rather than a figure.”
>
> Thank you for the suggestion, we will edit this in the final version.

---

> > ### Comment · Reviewer_gj2Y · 2025-08-04
> >
> > Thank you for all your explanation, which makes this work much clear to me. I would like to keep the positive recommendation.

---

> > > ### Author Response · Authors · 2025-08-05
> > >
> > > We're happy to hear that these explanations help clarify the work. We would appreciate you considering updating your rating if you feel the changes improve the final version. Thank you for your response.

---

### Official Review · Reviewer_JDeS · 2025-07-03

**Clarity:** 3
**Significance:** 3
**Originality:** 3
**Rating:** 4
**Confidence:** 3

**Summary:**

This paper introduces a new framework, Rationality-Preserving Policy Optimization (RPO), to address the problem of self-sabotage in adversarial optimization in multi-agent reinforcement learning (MARL). The authors propose the Rational Policy Gradient (RPG) algorithm to enforce rationality constraints via the introduction of manipulator agents. They demonstrate the framework’s ability to improve robustness, avoid sabotage, and produce diverse policies in cooperative and general-sum environments.

**Questions:**

Are manipulators optimized simultaneously with base agents, or is there an alternating schedule? Is there any stability concern when both influence each other via nested gradients?

**Ethical Concerns:**

["NO or VERY MINOR ethics concerns only"]

**Final Justification:**

After reading the rebuttal and other reviewers' comments, I think this paper needs further improvement for an acceptance. Therefore, I will keep my current rating.

**Quality:**

3

**Strengths And Weaknesses:**

Strengths:

The paper addresses an important and understudied failure mode (self-sabotage) in adversarial MARL.

The introduction of manipulator agents and rationality constraints is novel and shows potential in expanding the utility of adversarial training beyond zero-sum games.

Experimental results are comprehensive and support the claimed improvements in robustness, diversity, and stability.

Weaknesses:

While the paper provides strong motivation and intuitive examples (e.g., Figure 2), the mathematical formulation of the core problem is vague. For instance, the exact nature of the adversarial optimization problem being generalized is not clearly defined (e.g., is it a Markov game? Is the environment shared or decentralized?). The rationality constraint (Definition 3.1) is stated informally as “πi ∈ BR(π−i′)”, but the space of allowable π−i′ is not precisely defined. Are these stationary policies, stochastic, or deterministic?

---

> ### Author Rebuttal · Authors · 2025-07-31
>
> Thank you for your review. We appreciate your recognition that we “address an important and understudied failure mode” with “comprehensive experiments.” We address each point individually:
>
> > “the mathematical formulation of the core problem is vague. For instance, the exact nature of the adversarial optimization problem being generalized is not clearly defined (e.g., is it a Markov game? Is the environment shared or decentralized?). The rationality constraint (Definition 3.1) is stated informally as “πi ∈ BR(π−i′)”, but the space of allowable π−i′ is not precisely defined. Are these stationary policies, stochastic, or deterministic?”
>
> Section 2 contains some of these details: the problem is a general-sum partially-observable stochastic game (also known as a Markov game); the observations can be shared, decentralized, or partially shared according to the observation function O : S × A × Ω → [0, 1]; and the policies are stochastic.  We agree that these points are important and will be made more clear in the final version. The space of allowable π−i′ is the entire space Π−i and we will add this to the final version. Thank you for identifying these points as this will make the paper easier to understand.
>
> > “Are manipulators optimized simultaneously with base agents, or is there an alternating schedule? Is there any stability concern when both influence each other via nested gradients?”
>
> The manipulators are indeed optimized simultaneously with the base agents. The updates in equations (3) and (4) happen together on each time step. Alternating or nested optimization schedules is a very interesting direction for future research. Stability concerns are often an issue in multi-agent learning settings and we did observe oscillating behaviors in certain matrix games that are known to have unstable equilibrium (Figure 13). However, increasing the number of lookahead steps in RPG actually leads to stabilizing this oscillating behavior (also in Figure 13). Further exploring the ability for manipulators to stabilize multi-agent learning dynamics would be another exciting direction for future research.

---

> > ### Comment · Reviewer_JDeS · 2025-08-04
> >
> > Thank you for the clarification. Please include these in the final version.

---

> > > ### Author Response · Authors · 2025-08-05
> > >
> > > We will certainly include these clarifications in the final version. Please consider updating your rating if you feel the clarifications improve the final version. Thank you for your response and for taking the time to review our rebuttal and please let us know if you have any remaining concerns.

---

### Official Review · Reviewer_XfB2 · 2025-07-03

**Clarity:** 3
**Significance:** 3
**Originality:** 2
**Rating:** 4
**Confidence:** 4

**Summary:**

The paper shows that naïve adversarial optimisation in cooperative/general-sum MARL provokes irrational self-sabotage. It formalises Rationality-preserving Policy Optimisation (RPO) and instantiates it with Rational Policy Gradient (RPG), which inserts a “manipulator” agent and higher-order opponent-shaping gradients so each learned policy remains a best-response to some partner. RPG variants of AP, AT, PAIRED and AD are evaluated on matrix games, STORM, Overcooked and Hanabi, demonstrating (i) avoidance of sabotage, (ii) higher cross-play robustness, and (iii) discovery of genuine policy diversity.

**Questions:**

What is “self-sabotage”, can you give a formal definition / theorem? It was first documented in Cui et al 2023 but I think it was just described instead of officially formalized. Moreover, the self-sabotage issue described in this paper and Cui et al 2023 can be easily patched by masking out invalid actions with large negative values?

**Ethical Concerns:**

["NO or VERY MINOR ethics concerns only"]

**Limitations:**

1. The sample variance grows with look-ahead steps.
2. Experiments do not cover >2 agents or partially-observable continuous control, so generalizability is uncertain.

**Quality:**

3

**Strengths And Weaknesses:**

**Strength**

- This paper is well motivated to analyze and handle self-sabotage in diverse teammate generation problem, which has been a hard problem to define and tackle.
- Provides a principled optimisation formalism (RPO) plus a general recipe (RPG) that retrofits several prior algorithms. Conceptually this algorithm is generalizable to any reinforcement learning algorithms.
- Extensive experiments across four benchmarks; shows superior robustness/diversity versus strong baselines (SP, AD, CoMeDi, etc.). Ablations (look-ahead depth, partner-play coeff.) give insight into stabilizing dynamics.

**Weakness**

- Experiments remain in small/2-agent domains, although the formulation is pretty generalized.
- Theoretical analysis is insufficient:
    - L123, BR is a policy right? BR = \arg\max U(\pi_{i}, \pi_{-i}) for rigorousness
    - No clear explanation on O_i and L_dice in this paper, even though it is referred to related work.
    - Theoretically unclear why the method can prevent self-sabotage, and the paper does not define self-sabotage formally. Significant work should put into section 3 or 4 to explain the design of the method.
    - The method looks pretty like a fixed-size population version of CoMeDi combined with DiCE update. So I am not sure how the authors compare a fixed-size population method with an iterative-expanding population method to make sure it is a fair comparison.
- A few more related works on adversarial diversity & self-sabotage:
    - (This paper has a study on how to design adversarial diversity to reduce self-sabotage) Charakorn, Rujikorn, Poramate Manoonpong, and Nat Dilokthanakul. "Generating diverse cooperative agents by learning incompatible policies." *The Eleventh International Conference on Learning Representations*. 2023.
    - (This paper uses Lagrange constraint satisfaction to anneal the self-sabotage) Rahman, M., Cui, J., & Stone, P. (2024, March). Minimum coverage sets for training robust ad hoc teamwork agents. In *Proceedings of the AAAI Conference on Artificial Intelligence* (Vol. 38, No. 16, pp. 17523-17530).

---

> ### Author Rebuttal · Authors · 2025-07-31
>
> Thank you for your review. We are glad you found the work “well‑motivated” with “extensive experiments” and a “principled optimisation formalism.” We address each point individually:
>
> > “Experiments remain in small/2-agent domains, although the formulation is pretty generalized.”
>
> Our experiments use the same environments as many recent related research. Overcooked and Hanabi are regularly used to test various SOTA algorithms for cooperative environments. Indeed, Hanabi was identified as a challenge domain (Bard, Nolan, et al. "The hanabi challenge: A new frontier for AI research.") that still remains unsolved.
>
>
> > “L123, BR is a policy right? BR = \arg\max U(\pi_{i}, \pi_{-i}) for rigorousness?”
>
> Thank you for catching this – this is a typo and will be fixed in the final version.
>
>
> > “No clear explanation on O_i and L_dice in this paper …”
>
> O_i is the optimization objective for agent i. For example, in the case of adversarial training, for agent adversary,  O_adversary = min_{π_adversary} U_victim (π_victim , π_adversary). It is first introduced in section 3.2 but we will add an explanation of the notation to the preliminary section for clarity.
>
> L_dice is formally defined in the appendix in section E.1 in equation (10), but we agree that this is an important detail to include in the main section and will include it in Section 4 with the extra page allowed in the final version.
>
> > “Theoretically unclear why the method can prevent self‑sabotage, and the paper does not define self‑sabotage formally.”
> > What is “self-sabotage”, can you give a formal definition / theorem? It was first documented in Cui et al 2023 but I think it was just described instead of officially formalized.”
>
> Similar to existing work, we characterize self-sabotage as a general phenomenon when agents act against their own interests in order to minimize another player’s reward as part of an optimization problem. We do formally define Rationality-preserving Policy Optimization (Definition 3.1 on line 105) which we believe encapsulates the underlying issue behind self-sabotage by preventing agents from acting against their own interests.
>
> > “Moreover, the self-sabotage issue described in this paper and Cui et al 2023 can be easily patched by masking out invalid actions with large negative values?
>
> Unfortunately, the self-sabotage issue cannot simply be solved by masking out invalid actions. The sabotaging behavior is characterized as an entire state-action policy, involving a long sequence of actions and interactions that result in self-sabotage. It’s not clear how to precompute all of the policies that result in self-sabotaging behavior and it’s also not clear how to mask out this precomputed set of self-sabotaging policies during the optimization.
>
> > “The method looks pretty like a fixed-size population version of CoMeDi combined with DiCE update. So I am not sure how the authors compare a fixed-size population method with an iterative-expanding population method to make sure it is a fair comparison.”
>
> Our algorithm aims to solve the self-sabotage problem in a fundamentally different way than CoMeDi. While CoMeDi aims to limit the ability of agents to identify whether they are in self-play or cross-play during training, RPG entirely removes the incentive to self-sabotage. Appendix C includes a lengthy discussion of how RPG compares to existing approaches (including CoMeDi) and how our method overcomes the limitations of methods such as CoMeDi. Our paper demonstrates that CoMeDi fails to prevent self-sabotage in any of the environments and we don’t see how anything would improve for CoMeDi if it were changed to simultaneously learn the fixed-size population (as done with AT-RPG). Moreover, RPG is applicable to a wide range of adversarial optimization problems, whereas CoMeDi only applies to adversarial diversity.
>
> > “Sample variance grows with look‑ahead steps.”
>
> We are not sure this is necessarily true and would appreciate any more intuition behind this claim. In the limit of look-ahead steps the base agent converges to some minima which could end up being very similar behavior (i.e., low variance). Some long-term meta-gradient approaches take advantage of this property using the implicit function theorem (which we leave to future work). Existing meta-gradient approaches (that use lookahead) have scaled to large settings with long lookahead (50+ step lookahead in Engstrom, Logan, et al. "Optimizing ml training with metagradient descent."), so we believe there is a lot of potential for RPG to further scale.
>
> > “Related‑work omissions (Charakorn 2023, Rahman 2024).”
>
> Thank you for bringing these to our attention. We will include a discussion of these related works in the final version.

---

> > ### Comment · Reviewer_XfB2 · 2025-08-06
> >
> > >Similar to existing work, we characterize self-sabotage as a general phenomenon when agents act against their own interests in order to minimize another player’s reward as part of an optimization problem. We do formally define Rationality-preserving Policy Optimization (Definition 3.1 on line 105), which we believe encapsulates the underlying issue behind self-sabotage by preventing agents from acting against their own interests.
> >
> > It is just an objective for optimization, please formally define it ("self-sabotage"), with words or a theorem.

---

> > > ### Author Response · Authors · 2025-08-06
> > >
> > > We agree that introducing more formality to the concept of self-sabotage would help clarify our paper and strengthen our contributions. We will add a definition in Section 3.1 that formally defines self-sabotage in lines with the existing Definition 3.1. Here is a draft:
> > >
> > > > Optimization problems in multi-agent games often include objectives distinct from the incentives in the underlying game (i.e., agents maximizing their individual reward). When the optimization problem results in policies with *irrational behavior*, behavior that causes an agent to act against its own interests in the underlying game, we define this as *self-sabotage*.
> > >
> > > With this inclusion, it makes it clear how Rationality-preserving Policy Optimization (Definition 3.1) prevents self-sabotage. Thank you for the suggestion and helping improve the paper.

---

> ### Comment · Reviewer_XfB2 · 2025-08-08
>
> Thanks authors for providing this draft definition. I think the definition makes sense.
> Another suggestion is that your algorithm part might be missing an iteration loop through all players when updating $\theta'$, a mixture usgae of $k$ and $m$, and many narrations or paper organization could be clearer. Looking forward to the next revision of it.

---

### Official Review · Reviewer_9KSf · 2025-07-03

**Clarity:** 3
**Significance:** 3
**Originality:** 4
**Rating:** 6
**Confidence:** 3

**Summary:**

This paper presents a new formalism and method to enhance the adversarial training of methods in multi-agent reinforcement learning.
It targets the specific cooperative settings, where adversarial agents learning to find flaws in others' strategies can end up with a self-sabotage policy.
This implies that the adversarial agent ends up with a policy that prevents the victim from finding another policy that will improve its result.
This is also referred to as non-rational behavior.
The goal of RPO is to constrain the maximized adversarial objective by ensuring that the adversarial policy is the best response to one of the victim’s possible responses.
From this constraint, RPG is derived to improve different adversarial training algorithms by ensuring that no self-sabotage policy is achieved.
This new paradigm enables the development of more robust learning strategies.

The introduction reminds us of the context of adversarial training and its success in zero-sum games, inherent to the objective of each agent that benefits from the loss of others.
The problem of these approaches in a cooperative scenario is that agents usually learn a « self-sabotage » behavior, where the attacker agent does not cooperate anymore, and the victims are not able to find any policy leading to better performance.
This is claimed to be a non-rational behavior and the whole goal of the paper is to enforce rationality.
A clear example is provided showing the difference between a rational adversarial attack and a self-sabotage policy.
The introduction briefly describes how this rationality constraint is incorporated into the learning framework by introducing manipulators, copies of agents trained using opponent-shaping methods, thereby introducing an additional learning objective.
Contributions are: a new formalism (RPO), a new gradient-based method to train agents (RPG), five new adversarial algorithms extending existing ones with RPG, and an empirical demonstration of the benefits of the new paradigm.

Preliminaries define the stochastic game framework, where the objective of agents is to maximize the expected sum of discounted rewards dependent on the joint policy of all agents, and what constitutes a best response policy.

RPO is then defined.
A reminder of the adversarial optimization, its application in a simple cooperative game, and what is an irrational self-sabotage policy, and what is a rational one.
The Rational-preserving Policy Optimization is defined: for each agent i, its adversarial objective is maximized by pi_i while there must exist a joint-policy of the others, pi_-i, for which pi_i is the best response.
Its implication for a simple cooperative matrix game is provided to help build intuition.
Adversarial training RPO is thus a generalization of adversarial training in zero-sum games.

RPG is then defined and provides a solution to incorporate the RPO constraint into existing learning algorithms.
For each policy pi_i, there is a manipulator pi^M_i.
pi_i maximise the classical objective U(pi_i, pi^M_-i), enforcing pi_i to be the best response to pi^M_-i.
pi^M_-i, in parallel, are trained to maximise the adversarial objective.
How gradients are computed, and in which order, is then described, followed by a statement on partner-play regularization.
A representation of the algorithm is presented and then described.
Finally, the application of RPG to five existing adversarial algorithms is described.

Experiments demonstrate four claims clearly identified: RPG allows learning 1) diverse policies, 2) robust policies, 3) non-trivial adversarial examples, and 4) non-self-sabotaging policies.
Experiments are conducted in four environments: matrix games, Overcooked, STORM, and Hanabi, all of which are 2-player cooperative games (?).
Environments and parameters are detailed in the appendix.
The four claims are demonstrated through dedicated experiments and metrics, showcasing the power of RPG against six other methods, while applying RPG to four of them.

The paper finishes with a related work section, reminding the need to find a non-sabotage adversarial training method, claiming that to the best of their knowledge, RPG is the first solution using opponent-shaping to achieve adversarial training.
A future work direction is proposed: to use a line of work improving opponent-shaping with a more effective method in RPG.

**Questions:**

While RPO and RPG are defined for n-player stochastic games, experiments are two-player ones.
Linked to the clarity issues of the three different joint-policies mathematical expression, wouldn’t it be clearer only to use i and -i. And maybe to define everything in a two-player game and state that it generalizes to n-player ones. Especially because pi_-i, the co-player policy, can represent a single or a joint policy without problem in the definition.

What is the overhead of duplicating all agents and training several policies at the same time?
I’d be happy to have an incentive for the computational cost of the approach and its scalability with more than two agents.

At the beginning of the paper, zero-sum games are not considered because self-sabotage has no meaning in that setting.
However, this method could also improve the learning of policies in a competitive setting, especially when considering the first three claims empirically demonstrated.
Did you try the algorithm in a competitive setting?

**Ethical Concerns:**

["NO or VERY MINOR ethics concerns only"]

**Final Justification:**

I think this paper is a must for the community.
It is crystal clear that this new method can benefit the cooperative MARL community.
The number of environments tested is significant and the results benefit from a rigorous experimental protocol.
The authors addressed all my comments that will improve the paper.

**Limitations:**

The paper lacks a proper conclusion that summarizes the findings and nuances the results that RPG and its application to the five algorithms provide.

Suggestion: line 35, I would add the statement of line 99: preventing meaningful learning, because there exists no policy that will improve the returns of the attacked teammate.

Suggestion: Implicitly, the Hanabi games tested are two-player, but it should be stated explicitly.

Typo line 160 peusocode

**Quality:**

4

**Strengths And Weaknesses:**

The paper provides a high-quality explanation of the problem and the solution, and appears to be the first to address it, supported by rigorously conducted experiments. However, some clarity enhancement could benefit the reader, and a consideration of limitations and extensions beyond two-player cooperative settings could be added.

Quality:

(+) The submission is technically sound, and all claims are well supported with examples, references, and experiments.

(+) The paper is standalone, and all details are provided in the appendix.

(+) Experiments are rigorous, and many details are provided to analyse experiments, supporting the claims.

Clarity:

(+) The paper is overall clearly written

(-) I believe the three joint policies (pi_adversary, pi_victim), (pi_i, pi_-i) , (pi_1,…, pi_m) are sometimes used interchangeably, making Section 3.2, among others, hard to follow. While I believe it clarifies section 3.1, using all three of them is overwhelming.


Significance:

(+) This paper has a significant impact on the community, as it presents the first method to address the self-sabotage problem in adversarial training.

(+) Many existing methods are mentioned, tested, and surpassed by RPG.

Originality:

(+) Along with possibly improving some readers’ understanding of self-sabotage, the approach is clearly original in solving it.

---

> ### Author Rebuttal · Authors · 2025-07-31
>
> Thank you for your review. We appreciate your noting that our “paper has a significant impact on the community” and that the experiments are “rigorously conducted.” We address each point individually:
>
> > “(-) I believe the three joint policies (pi_adversary, pi_victim), (pi_i, pi_-i) , (pi_1,…, pi_m) are sometimes used interchangeably, making Section 3.2, among others, hard to follow. While I believe it clarifies section 3.1, using all three of them is overwhelming.”
>
> > “While RPO and RPG are defined for n-player stochastic games, experiments are two-player ones. Linked to the clarity issues of the three different joint-policies mathematical expression, wouldn’t it be clearer only to use i and -i. And maybe to define everything in a two-player game and state that it generalizes to n-player ones.  Especially because pi_-i, the co-player policy, can represent a single or a joint policy without problem in the definition.”
>
> Thank you for this feedback – we agree that the amount of notation through section 3 is overwhelming and we will improve this in the final version. We agree that it would be most clear to simply use i and -i in as many places as possible (potentially simply eliminating pi_adversary and pi_victim). However, the reason we also use the (pi_1,…, pi_m) is because even while the underlying game might only involve two players, the optimization problem on top of that game could involve more than two policies. For example, the adversarial diversity algorithm aims to find a set of N pairs of policies that are as diverse (i.e. incompatible) as possible. The choice of N can be larger than 2. We will do a better job of clarifying this distinction in Section 3 and simplify the notation where possible.
>
> > “What is the overhead of duplicating all agents and training several policies at the same time? I’d be happy to have an incentive for the computational cost of the approach and its scalability with more than two agents.”
>
> We agree that this is an important detail to add to the paper and will add it to the final version for each of our experiments. On average, the original adversarial diversity algorithm with a population size of 2 is 2 times slower in wall-clock time (per step) than self-play while the AD-RPG (ours) is an additional 3 times slower than AD (due to the extra cost of manipulator policies and computing higher-order gradients). The computational overhead of the higher-order gradients is constant in the number of agents so the method should scale well beyond two agents. However, we will leave this investigation for future work. Moreover, recent works have demonstrated paths for further scaling higher-order gradients (Logan, et al. "Optimizing ml training with metagradient descent").
>
> > “At the beginning of the paper, zero-sum games are not considered because self-sabotage has no meaning in that setting. However, this method could also improve the learning of policies in a competitive setting, especially when considering the first three claims empirically demonstrated. Did you try the algorithm in a competitive setting?”
>
> The main reason we did not test our algorithm in a zero-sum setting is because RPO is a strict-generalization of the associated algorithms from the zero-sum setting (explained on lines 121-124 of the submission) since adversarial incentives (minimizing your opponent’s reward) and individual incentives (maximize your own score) coincide. For example, AT-RPO simply reduces to the same problem as AT in the zero-sum setting. Because of this, we believed that using RPG in the zero-sum setting would just add unnecessary complexity to the algorithm. However, inspired by your question, we decided to empirically test our implementation of RPG in a zero-sum setting. We picked the matrix game of rock-paper-scissors:
>
> |       | **R**   | **P**   | **S**   |
> | ----- | ------- | ------- | ------- |
> | **R** | (0, 0)  | (-1, 1) | (1, -1) |
> | **P** | (1, -1) | (0, 0)  | (-1, 1) |
> | **S** | (-1, 1) | (1, -1) | (0, 0)  |
>
> and tested both AT and AT-RPG. Both methods have very similar behavior in which the adversary and victim oscillate between picking rock, paper, or scissors during training (similar to the oscillating behavior in Figure 13). We found that increasing the lookahead (e.g. greater than 8) in AT-RPG led to this behavior stabilizing and converging on playing the strategy that randomizes across the three actions (in the same way lookahead stabilizes the oscillations in Figure 13). Thank you for asking this question – we will include these results in the final-version and believe it will strengthen the paper.
>
> > “The paper lacks a proper conclusion …”
>
> We will add a proper conclusion with the extra page allowed for the camera-ready version. Here is a draft:
>
> We introduce the problem of *rationality-preserving policy optimization* and give a solution in the form of a gradient-based algorithm called *rational policy gradient (RPG)*. We demonstrate the diverse uses of RPG by constructing algorithms that robustify agents, expose rational adversarial examples, and learn genuinely diverse behaviors across general-sum settings. Our results highlight that the success of adversarial optimization in zero-sum training methods can be extended to the general-sum setting and we aim to inspire further research in this direction. Furthermore, we hope that the generality of RPG will enable the development of entirely new classes of adversarial optimization algorithms in general-sum settings.
>
> > “Implicitly, the Hanabi games tested are two‑player, but it should be stated explicitly.”
>
>  We will state this in section 5.4 and the caption of Fig 7.
>
> > “Typo line 160 peusocode” and “Line 35”
>
>  Thank you for the clarification suggestion – the typo will be fixed, and the line‑35 clarification added.

---

> > ### Comment · Reviewer_9KSf · 2025-08-04
> >
> > Thanks for the clarification and the additional content provided to the paper.
> > I am happy to increase my score.

---

> > > ### Author Response · Authors · 2025-08-05
> > >
> > > Thank you for reviewing our rebuttal and helping improve the paper with your feedback. We appreciate you increasing your score.

---

### Note · Authors · 2025-08-12

Dear Reviewers, AC, and SAC,

We sincerely thank you for the effort put into your reviews along with your constructive comments and feedback. We believe the final version of the paper will be significantly strengthened as a result. We wanted to briefly highlight a few of the improvements that will be included in the final version of the paper as a result of our discussion:
- **Additional experiments:** a new set of experiments testing our algorithm in zero-sum settings, backing up our claim that RPG strictly generalizes adversarial algorithms from the zero-sum setting. We have also included additional details about the computation cost of the variations of RPG algorithms.
- **A formal definition of self-sabotage:** this definition adds precision to a loosely defined concept from prior literature.
- **Addition of a conclusion section:** we have included a proper conclusion section to highlight key takeaways from the paper.
- **Clarification throughout:** notation in Section 3.1; explanations of O_i and L_dice; distinction from existing algorithms (e.g. CoMeDi); discussion of additional related works; improved narrations; clarification of Claim 3 and Figure 7; and much more.

---

### Decision · Program_Chairs · 2025-09-17

**Decision:**

Accept (poster)

**Comment:**

This paper introduces Rationality-preserving Policy Optimization (RPO), a framework that overcomes the self-sabotage problem that arises when applying adversarial optimization to cooperative multi-agent settings by ensuring that agents remain rational---i.e., that they adopt a best response policy to the policies of possible partner agents. The authors propose a gradient-based algorithm that solves RPO, called Rational Policy Gradient (RPG). The paper is well-motivated and addresses an important problem in multi-agent RL. Both the framework and the proposed technique appear novel, although they resemble some methods proposed in prior work. While some reviewers find the empirical evaluation comprehensive, others feel it could be strengthened with additional complex multi-agent environments. That said, the approach is empirically evaluated using standard benchmarks for cooperative multi-agent RL. Some aspects of the paper, including the formal setting, could be further clarified in the final version, and the authors are encouraged to do so.